# Post-pandemic changes in population immunity have reduced the likelihood of emergence of zoonotic coronaviruses

Ryan M. Imrie [1,7], Laura A. Bissett[1,2,7], Savitha Raveendran[1], Maria Manali[1], Julien A. R. Amat[3], Laura Mojsiejczuk[1], Nicola Logan [1], Andrew Park [4], Marc Baguelin[5,6], Mafalda Viana [2] ✉, Brian J. Willett [1] ✉ & Pablo R. Murcia [1] ✉

Infections by endemic viruses, and the vaccines used to control them, often provide cross-protection against related viruses, potentially altering the transmission dynamics and likelihood of emergence of new zoonotic viruses with pandemic potential. Here, we investigate how population immunity after the COVID-19 pandemic has impacted the likelihood of emergence of a novel sarbecovirus, termed SARS-CoV-X. To this end, we combined empirical cross-neutralisation data with mathematical modelling to identify key immunological and epidemiological factors shaping sarbecovirus emergence. We show that sera from individuals with different COVID-19 immunological histories contained cross-neutralising antibodies against the spike (S) protein of multiple zoonotic sarbecoviruses. Simulations parameterised by these data predict that the likelihood of emergence of a novel sarbecovirus has been reduced significantly by population cross-immunity, with outcomes determined by the extent of cross-protection and R0 of the novel virus. Preventative vaccination against SARS-CoV-X using available COVID-19 vaccines can help resist emergence even in the presence of co-circulating SARS-CoV-2. However, a theoretical vaccine with high specificity to SARS-CoV-2 can increase emergence probability by suppressing SARS-CoV-2 prevalence and, by extension, levels of natural cross-protection. Overall, SARS-CoV-2 circulation and vaccination have generated widespread immunity against related sarbecoviruses, creating an immunological barrier to novel sarbecovirus emergence in humans.

Identifying viruses with zoonotic potential before they emerge is essential for pandemic preparedness. To infect humans, zoonotic viruses must overcome multiple biological barriers. As intracellular pathogens, viruses rely on host machinery for replication, but molecular incompatibilities with host proteins can restrict infection. Humans have antiviral defenses, including restriction factors[1], immune cells such as natural killer (NK) cells[2] and cytotoxic T cells (CTLs)[3], and interferons that suppress replication and promote immunity[4]. Neutralizing antibodies, induced by infection or vaccination, further limit replication and aid recovery[5]. As sustained human-to-human transmission is a prerequisite for viral emergence, not all zoonotic viruses that overcome these barriers pose a pandemic risk, and the risk of emergence by viruses that are either non-transmissible or have limited onward transmissibility is considered low[6].

Three highly pathogenic coronaviruses—SARS-CoV, MERS-CoV, and SARS-CoV-2—have crossed from animals to humans in the 21st century[7]. While human-to-human transmission has been reported for all of them, their epidemiological outcomes differed. SARS-CoV emerged in China in 2002, caused an international outbreak with 8096 cases and 774 deaths[8] and was controlled by 2003, with only five further zoonotic infections reported[9]. MERS-CoV causes sporadic zoonotic infections with limited spread[10]; as of October 2025, 2640 cases have been confirmed, with the largest outbreak reaching 186 cases[11,12]. In contrast, SARS-CoV-2 emerged in late 2019, leading to a global pandemic with over 770 million cases and more than 7 million deaths[13]. Within a year, vaccines were developed and deployed globally, with over 13 billion doses administered to date[14].

Mutations in the spike (S) protein of SARS-CoV-2 have enabled the virus to escape immunity induced by vaccination or infection, resulting in epidemic waves where reinfections and vaccine breakdowns are common[15]. SARS-CoV-2 continues to circulate even in populations with high vaccination coverage[16], although hospitalizations and deaths have decreased significantly compared to the pre-vaccine era[17]. As a result, the global immunological landscape has changed dramatically since the emergence of COVID-19, as a large proportion of the human population has developed anti-SARS-CoV-2 antibodies. For example, in Scotland alone, it was estimated that the proportion of SARS-CoV-2 seropositive individuals increased from 4.4% in May 2020 to 96.5% in June 2022[18].

Cross-immunity can impact the transmission dynamics of genetically related respiratory viruses, as illustrated by the dynamic patterns of human respiratory syncytial virus, metapneumovirus, parainfluenza viruses and enteroviruses[19,20]. While recent reports explored the effectiveness of SARS-CoV-2 immunity against infection, hospitalization, or death by different SARS-CoV-2 variants[21,22], no studies have investigated the impact of SARS-CoV-2 cross-immunity on the likelihood of emergence of novel zoonotic viruses. Various surveillance and virus discovery studies have shown that viruses belonging to the same subgenus as SARS-CoV and SARS-CoV-2 (sarbecoviruses) circulate in wildlife over broad geographic regions[23,24]. Examples include the bat viruses Rs4084, RaTG13, BANAL-52, BANAL-103, BANAL-114, BANAL-236, BANAL-247 (all the BANAL CoVs can bind the human ACE2 receptor)[25], as well as the pangolin viruses GD-1, GX-P5L and GX/P1E[26].

A key goal of pandemic preparedness is to assess the risk posed by animal-origin viruses[27], including zoonotic sarbecoviruses—collectively referred to here as "SARS-CoV-X". We hypothesized that post-pandemic immunity has reduced the likelihood of antigenically related sarbecoviruses emerging in humans. To test this, we measured the extent to which SARS-CoV-2 antibodies neutralize related viruses using sera from four groups: unexposed individuals; those who had recovered from infection; vaccinated individuals; and those with immunity from both infection and vaccination (i.e., hybrid immunity). The sarbecovirus panel included two close relatives of SARS-CoV-2—the bat virus RaTG13 and the pangolin virus GX/P1E—as well as SARS-CoV and its close relative, the bat virus RS4084. We then built an age-stratified stochastic SEIRS model, based on the Scottish population, to simulate co-circulation of SARS-CoV-2 and the hypothetical SARS-CoV-X. The model varied vaccination uptake, cross-immunity, and SARS-CoV-X transmissibility ($R_0$) to estimate how existing immunity reduces emergence risk. Additionally, as virus surveillance efforts aim to act as an early-warning system for potential emergence events[28], and widespread vaccination with a cross-reactive vaccine is a possible preventative action[29], we explored how a 2-month preventative vaccination program using available COVID-19 vaccines could impact the likelihood of sarbecovirus emergence in the presence of SARS-CoV-2 co-circulation. These simulations provide a conceptual exploration of how population immunity, vaccination, and transmissibility interact to influence virus emergence.

## Results

### SARS-CoV-2 infection and vaccination produce cross-reactive antibodies against other zoonotic sarbecoviruses

Neutralising activity in sera from individuals with different SARS-CoV-2 infection and vaccination histories was quantified in vitro using lentiviral pseudotypes expressing luciferase and the spike proteins of four sarbecoviruses: SARS-CoV, Rs4084 (bat), GX/P1E (pangolin), and RaTG13 (bat) (Fig. 1A-C). Sera from naïve individuals showed the lowest levels of neutralisation across viruses ($\mu = 14.8\%$, SEM = 11.2%). Neutralising activity was significantly higher in those with prior infection ($\mu = 41.7\%$, SEM = 10.8%, $p < 0.001$); those vaccinated once ($\mu = 45.2\%$, SEM = 11.3%, $p < 0.001$); and vaccinated twice (mean = 54.5%, SEM = 11.2%, $p < 0.001$). Neutralising activity increased further in individuals with hybrid immunity who had received one ($\mu = 64.1\%$, SEM = 10.9%, $p < 0.001$) and two vaccine doses ($\mu = 67.0\%$, SEM = 10.8%, $p < 0.001$) (Supplementary Tables 1–2). Cross-neutralisation also varied between viruses, with the lowest value across immune groups seen for SARS-CoV ($\mu = 29.9\%$, SEM = 13.2%). Significant increases were observed for Rs4084 ($\mu = 39.7\%$, SEM = 13.2%, $p < 0.001$), GX/P1E ($\mu = 43.2\%$, SEM = 13.2%, $p < 0.001$), and RaTG13 ($\mu = 78.6\%$, SEM = 13.2%, $p < 0.001$) (Supplementary Tables 3-4). These values are consistent with a decrease in cross-neutralisation with increasing evolutionary distance from the SARS-CoV-2 (Wuhan-Hu-1) spike protein (Fig. 1D). Accordingly, a significant positive relationship between spike protein amino acid similarity (%) and cross-neutralisation was detected ($\beta^2 = 0.09$, SEM = 0.003, $p < 0.001$) (Fig. 1E, Supplementary Tables 5–6). These results indicate that antibodies elicited against SARS-CoV-2 can cross-neutralise other sarbecoviruses, with levels influenced by the route of immune acquisition (natural infection, vaccination, or hybrid) and spike protein sequence similarity.

### A conceptual epidemiological model of SARS-CoV-2 and SARS-CoV-X co-circulation

To assess the risk of zoonotic sarbecovirus emergence—defined here as a virus reaching endemicity after exposure to the human population—under varying levels of cross-protection, vaccination, and $R_0$, we developed an age-stratified stochastic SEIRS model parameterised to resemble the vaccination status, SARS-CoV-2 trajectory, and projected demographic structure of Scotland from 2020-2028 (Supplementary Table 7). The model includes separate EIR compartments for the SARS-CoV-2 variants (Wuhan, Alpha, Delta, Omicron) and SARS-CoV-X, for each immune group defined by vaccination and infection history from the pseudotype neutralization data (Fig. 2A-B). To make the conversion from in vitro pseudotype neutralization to in vivo protection from infection, we have assumed that transmission probabilities scale by $(1 - x)$ of cross-neutralisation, linearly reducing infection risk in individuals with higher immune cross-reactivity. Scottish COVID-19 vaccination rates, including spring/winter booster programs, along with waning immunity calculated from the half-lives of vaccine-derived antibodies, produced a dynamic immune landscape of vaccine protection and cross-protection, peaking in winter with ~12–19% of the population expressing a vaccine-protective phenotype, dominated by individuals aged 65+ (Supplementary Fig. 1).

To provide estimates for $R_0$, the recovery rate, and immune waning rates for each SARS-CoV-2 variant, which vary considerably across the available literature, we calibrated our model using an approximate-Bayesian computation (ABC) process, fitted to prevalence data from the Office for National Statistics (ONS) Infection Survey Dataset[30] (see Supplementary Table 8 for prior and approximate posterior summaries). The model closely tracked real-world SARS-CoV-2 prevalence until the initial Omicron wave (Fig. 2C), after which serological, vaccination, and contact data became more limited. Without this information, the additional complexity of modelling identical successive immune-escape Omicron lineages was not justified, and SARS-CoV-2 prevalence instead stabilises at 2.2%,

 

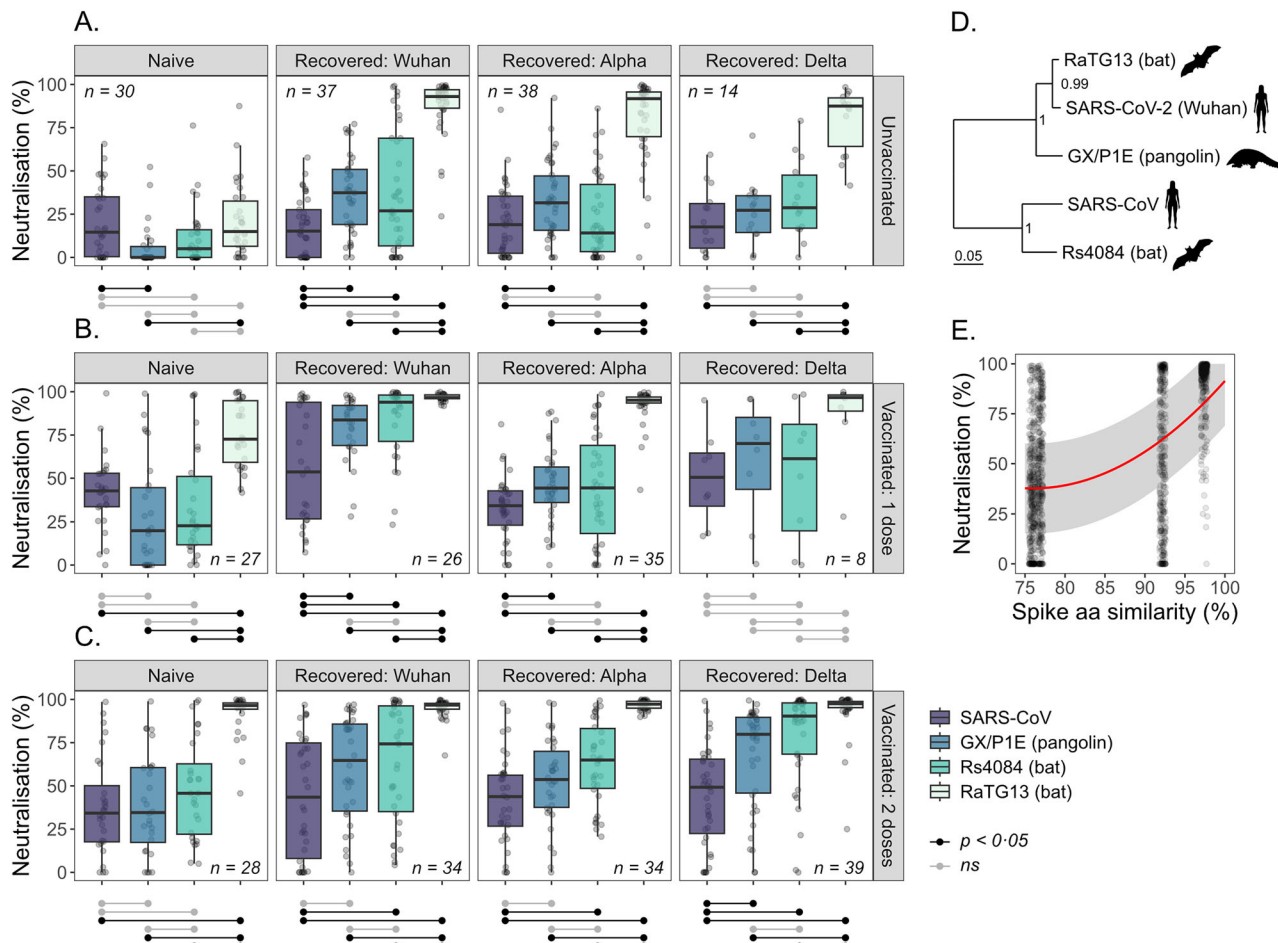

**Fig. 1 | Neutralisation of viral pseudotypes carrying the spike proteins of different sarbecoviruses by sera from individuals of different infection and vaccination histories.** Boxplots show the percentage neutralisation of viral pseudotypes by sera from individuals who were unvaccinated (**A**), vaccinated once (**B**), or vaccinated twice (**C**) against SARS-CoV-2. Results are separated into subplots (columns) based on an individual's history of natural infection with SARS-CoV-2, and separate boxplots are shown for neutralisation of viral pseudotypes carrying the spike protein of SARS-CoV (purple), Rs4084 (blue), GX/P1E (cyan) or RaTG13 (off-white). Boxplots show the median (centre line), interquartile range (box; 25th–75th percentiles), and whiskers extending to 1.5x the interquartile range. Sample sizes (*n*) indicate the number of unique patient sera for each combination of natural infection and vaccination history. Significant differences in neutralising activity are shown with black horizontal lines below each subplot, assessed using two-sided Welch's t-tests with Holm correction for multiple testing. **D** A maximum clade credibility phylogeny inferred from the spike protein sequences of each SARS coronavirus and SARS-CoV-2 (Wuhan-Hu-1), with posterior probabilities of each clade and a scale bar demonstrating amino acid substitutions per site per unit time. **E** The relationship between percentage neutralisation and spike protein sequence similarity (%) to SARS-CoV-2 (Wuhan-Hu-1) modelled as a quadratic function. Data points are neutralisation data taken from (**A**–**C** (excluding Naive + Unvaccinated individuals). The trend line and shaded area indicate the predicted values and standard error, respectively, from a quadratic mixed-effects model, with random effects of vaccine history, natural infection history, and serum ID.

corresponding to the mean prevalence across the Omicron peaks, avoiding unsupported assumptions about short-term variant-specific dynamics.

### Population immunity against SARS-CoV-2 decreases the likelihood of emergence of zoonotic sarbecoviruses

The model assessed the risk of a novel sarbecovirus emerging in humans under two scenarios: a) ongoing SARS-CoV-2 circulation with existing COVID-19 vaccination levels, and b) a simulated 2-month vaccination campaign using current COVID-19 vaccines to control SARS-CoV-X emergence. In the second scenario, the vaccine campaign was allowed to vary in the level of vaccine uptake (Fig. 2D), the timing of the start of the campaign relative to the date of SARS-CoV-X detection (Fig. 2E) and in the amount of cross-immune protection the vaccine provides from SARS-CoV-X infection (Fig. 2F). As no $R_0$ estimates are available for the zoonotic sarbecoviruses in a comparable human population to the model, the $R_0$ of each SARS-CoV-X virus was set to the model-fitted estimate for the Wuhan strain in this population

($R_0 = 1.57$). Sobol sensitivity analysis, performed on parameters for the $R_0$ of SARS-CoV-2 and SARS-CoV-X; the levels of natural immune protection; the waning rates of vaccine and natural immunity; and the simulated vaccination campaign, showed that most of the variation in the probability of SARS-CoV-X emergence was contributed to by the levels of natural cross-immunity and the $R_0$ of SARS-CoV-X (Table 1). Weaker effects on emergence probability were found for the $R_0$ of SARS-CoV-2; the timing, coverage, and cross-protection of the preventative vaccine program; and the waning rate of SARS-CoV-X immunity, while no effect was detected for the waning rate of SARS-CoV-2 immunity across the parameter ranges tested.

Simulations suggested the sarbecoviruses tested in our neutralisation assays had a modest emergence probability in immunologically-naïve populations (0.203, 95% CI: 0.191–0.216), and that this probability has decreased under current conditions of population cross-immunity created by SARS-CoV-2 infection and seasonal vaccination (Supplementary Table 9). The highest probability under current conditions was observed for SARS-CoV (0.0702, 95% CI:

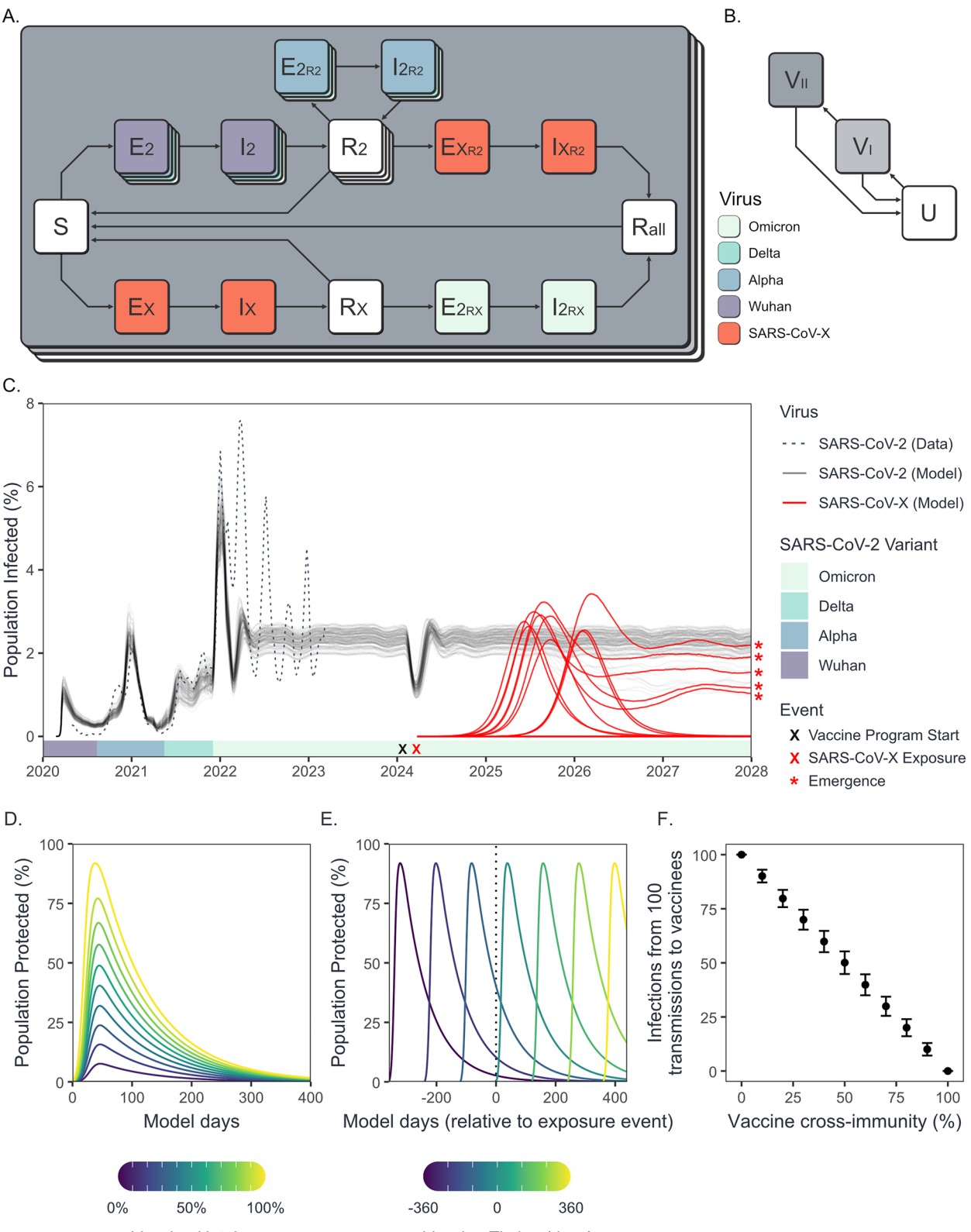

0.0699–0.0705), followed by Rs4084 (0.0632, 95% CI: 0.0629–0.0635), GX/P1E (0.0432, 95% CI: 0.0430–0.0435), and RatG13 (0.000025, 95% CI: 0.000019–0.000033). The preventative vaccination campaign had little effect on the probability of emergence when initiated more than 200 days before or after the first SARS-CoV-X case (Fig. 3, Supplementary Fig. 2). However, when vaccination began closer to the SARS-CoV-X exposure date, it significantly reduced emergence

probability. The most effective timing was at the point of SARS-CoV-X introduction, reducing emergence to 0.0352 (0.0343–0.0361) for SARS-CoV, 0.0248 (0.0239–0.0256) for Rs4084, and 0.0110 (0.0104–0.0117) for GX/P1E. These results show that short-term and widespread vaccination with currently available COVID-19 vaccines can reduce the emergence potential of novel sarbecoviruses, even alongside co-circulating SARS-CoV-2.

**Fig. 2 | An epidemiological model of SARS-CoV-2 and SARS-CoV-X co-circulation. A** The core structure of the model contains five unique EI compartment groups describing i) infection of naïve (S) individuals with SARS-CoV-2 (E2, I2); ii) infection of naïve individuals with SARS-CoV-X (EX, IX); iii) re-infection of individuals recovered from an earlier variant of SARS-CoV-2 (R2) with a later variant (E2R2, I2R2); iv) re-infection of individuals recovered from SARS-CoV-2 with SARS-CoV-X (EXR2 and IXR2); and v) re-infection of individuals recovered from SARS-CoV-X (RX) with SARS-CoV-2 (E2RX and I2RX). This structure is repeated across three layers representing different levels of vaccine protection against infection (**B**), where individuals may be unvaccinated (U), protected by a single dose (V$_I$), or protected by two doses (V$_{II}$). **C** An example 100-iteration model run showing percentages of individuals infected with SARS-CoV-2 (black) and SARS-CoV-X (red)

over time. The predominant SARS-CoV-2 variant is indicated by the shaded lower area, and estimates of real SARS-CoV-2 prevalence, taken from the ONS UK Coronavirus Infection Survey, are shown as a dotted line. In this example, preventative vaccination began 30 days before SARS-CoV-X emergence, with a vaccine uptake of 100%, and SARS-CoV-X phenotypes were set to those of SARS-CoV. During the preventative vaccine program, the percentage of individuals with vaccine-derived protection from infection over time is influenced by both the vaccine uptake (**D**) and vaccine timing (**E**) model parameters. **F** Points show the mean number of successful infections that occur from 100 transmissions to vaccinated individuals at different parameter values for vaccine-cross immunity. Error bars indicate the variability of outcomes (±1 standard deviation) arising from the underlying stochastic process.

**Table 1 | Sobol sensitivity analysis of SARS-CoV-X, SARS-CoV-2, and vaccine program parameters**

| Parameter | Distribution | First Order (95% CI) | Total Order (95% CI) |
|---|---|---|---|
| Natural cross-immunity | $U_{[0, 1]}$ | 0.469 (0.460, 0.477) | 0.524 (0.516, 0.532) |
| $R_0$ SARS-CoV-X | $U_{[2, 6]}$ | 0.417 (0.409, 0.425) | 0.454 (0.446, 0.462) |
| $R_0$ SARS-CoV-2 | $U_{[2, 6]}$ | 0.019 (0.016, 0.021) | 0.044 (0.042, 0.046) |
| Vaccine cross-immunity | $U_{[0, 1]}$ | 0.011 (0.010, 0.013) | 0.025 (0.023, 0.026) |
| Vaccine waning rate | $U_{[0, 0.02]}$ | 0.004 (0.003, 0.005) | 0.011 (0.010, 0.013) |
| Vaccine program start | $U_{[-90, 90]}$ | 0.003 (0.002, 0.004) | 0.011 (0.009, 0.012) |
| Vaccine program uptake | $U_{[0, 1]}$ | 0.002 (0.001, 0.002) | 0.005 (0.004, 0.006) |
| $R_{SARS-x}$ waning rate | $U_{[0, 0.02]}$ | 0.002 (0.001, 0.003) | 0.004 (0.003, 0.005) |
| $R_{SARS-2}$ waning rate | $U_{[0, 0.02]}$ | 0.001 (0.000, 0.002) | 0.003 (0.003, 0.004) |

First-order indices are estimates of the proportion of total variance in the probability of SARS-CoV-X emergence explained independently by each parameter, and total-order indices are estimates of the combined contribution of each parameter, including both its independent effect and all interaction effects. Values in parenthesis are 95% bootstrap confidence intervals based on 1000 bootstrap replicates and 1 million Monte Carlo samples in the Sobol design.

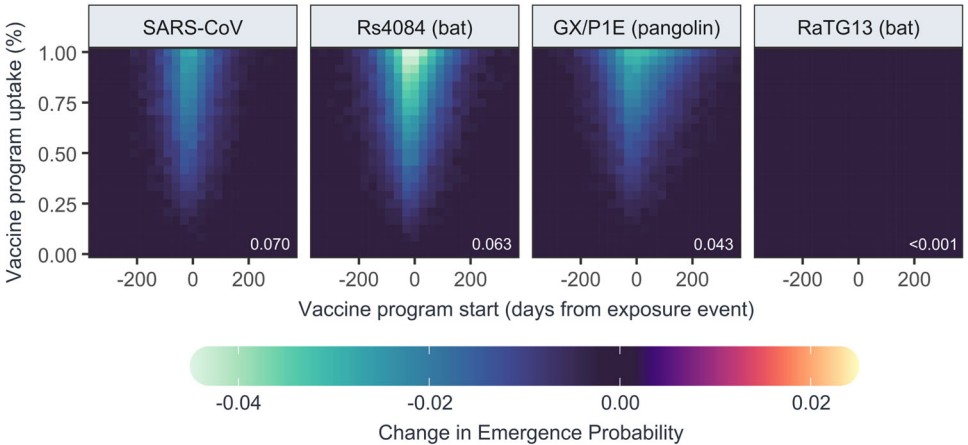

**Fig. 3 | Probability of emergence of different SARS coronaviruses in the presence of vaccination and co-circulating SARS-CoV-2.** Heatmaps show point estimates of the change in probability of emergence for four SARS coronaviruses in a population with co-circulating SARS-CoV-2 under varying conditions of

preventative vaccination, estimated from 100,000 model iterations for each unique combination of virus and vaccine program parameters. Inset white text shows the background probability of emergence for each virus.

To further characterise the conditions affecting SARS-CoV-X emergence, model runs were performed for hypothetical sarbecoviruses with varying $R_0$ values (2-6) and assuming equal levels of cross-protection from prior SARS-CoV-2 infection and vaccination (ranging from 0-100% protection from infection). Emergence probability increased with $R_0$ and decreased with stronger cross-immunity (Supplementary Fig. 3), such that a hypothetical sarbecovirus with no immune cross reactivity and an $R_0$ of 6 had an estimated emergence probability of 0.6108 (0.6096–0.6121), decreasing to 0.1799 (0.1789–0.1808) as the level of cross reactivity approached 100%.

Preventative vaccination campaigns were most effective when both vaccine cross-protection and SARS-CoV-X $R_0$ were high (Fig. 4, Supplementary Fig. 4), reducing the emergence probability in the above scenario from 0.1199 to 0.0106 (0.0094–0.0120) with 100% uptake and optimal timing. However, vaccination provided little additional benefit over the effect of naturally-acquired cross-immunity in cases where cross-immunity was high and SARS-CoV-X $R_0$ was low, with the effect of vaccination in these cases only evident in models where vaccine effectiveness varied independently from cross-immunity to SARS-CoV-2 (Supplementary Fig. 5).

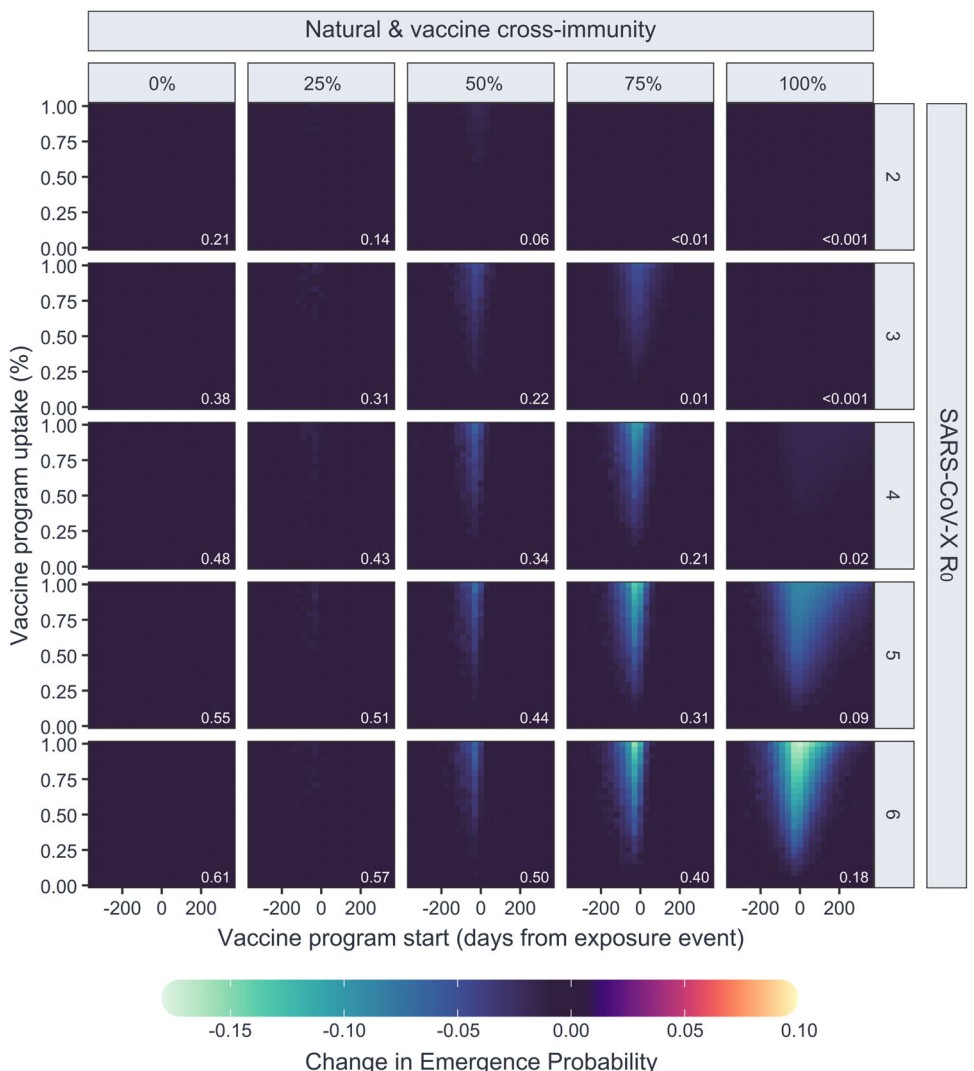

**Fig. 4 | Probability of emergence of theoretical SARS coronaviruses under conditions of equal natural and vaccine cross-immunity.** Heatmaps show point estimates of the change probability of emergence for 25 theoretical SARS coronaviruses with different $R_0$ values (panel rows) and levels of natural and vaccine-derived cross-immunity (panel columns) in a population with co-circulating SARS-CoV-2, estimated from 25,000 model iterations for each unique combination of virus and vaccine program parameters. In these scenarios, protection against SARS-CoV-X infection conferred from recovering from natural infection with SARS-CoV-2 ("natural cross-immunity") and vaccination ("vaccine cross-immunity") are identical. Inset white text shows the background probability of emergence for each virus.

## A vaccine with high SARS-CoV-2 specificity can increase the likelihood of SARS-CoV-X emergence

As cross-immunity from both vaccination and infection can decrease the likelihood of SARS-CoV-X emergence, it may be possible for a highly-specific SARS-CoV-2 vaccine with little cross-reactivity to increase the probability of SARS-CoV-X emergence, provided it reduces SARS-CoV-2 prevalence sufficiently to lower the level of natural cross-immunity in the population. This outcome was visible in our model when vaccine cross-reactivity was set very low (5%) (Fig. 5, Supplementary Fig. 6). The detrimental effect of vaccination under these conditions became more pronounced as SARS-CoV-X $R_0$ and natural cross-immunity increased. For a SARS-CoV-X virus with an $R_0$ of 6, this detrimental effect caused the probability of emergence to increase from 0.2197 (0.2186–0.2207) to 0.3250 (0.3192–0.3308) in the worst case. Even in the presence of detrimental vaccination, a virus with an $R_0$ of 2 remained incapable of emerging under conditions of 100% natural cross-immunity to SARS-CoV-2. However, in an additional run under conditions of 80% natural cross-immunity, vaccination with a 5% cross-reactive vaccine allowed a virus with no detectable ability to emerge in the human population to emerge in 2.36% (2.17–2.56%) of trials (Supplementary Fig. 7). The detrimental effect of vaccination decreases rapidly as the levels of vaccine-cross protection increase; in scenarios where vaccine-cross protection was set to one-third the effectiveness of natural cross-protection, the effects of the vaccine campaign on natural and vaccine-derived population immunity become effectively balanced, producing little detectable positive or negative effect of a preventative vaccination campaign on the probability of SARS-CoV-X emergence (Supplementary Fig. 8).

## Discussion

The emergence of COVID-19 highlights the serious threat of viral zoonoses and the need for effective ways to assess emergence risk and intervention impact. Here, we investigated the likelihood of a new zoonotic sarbecovirus emerging in the post-pandemic era—defined here as the virus reaching endemicity after exposure to the human population—given current levels of natural and vaccine-derived SARS-CoV-2 immunity. Our findings suggest that cross-reactive adaptive immunity offers substantial protection against sarbecovirus

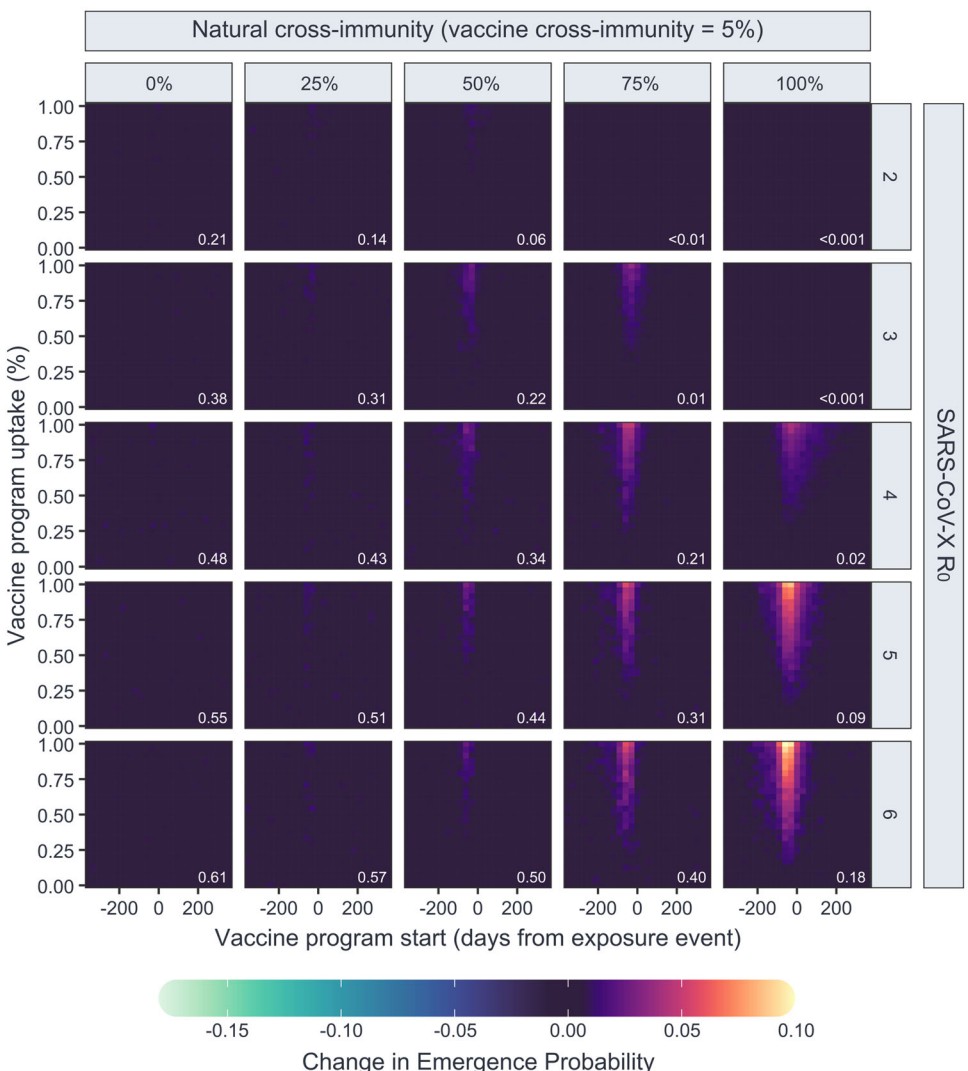

**Fig. 5 | Probability of emergence of theoretical SARS coronaviruses under conditions of low vaccine cross-immunity and high natural cross-immunity.** Heatmaps show point estimates of the change in probability of emergence for 25 theoretical SARS coronaviruses with different $R_0$ values (panel rows) and varying conditions of natural cross-immunity (panel columns) in a population with co-circulating SARS-CoV-2, under and low (5%) vaccine cross-immunity, estimated from 25,000 model iterations for each unique combination of virus and vaccine program parameters. Inset white text shows the background probability of emergence for each virus.

emergence, with close relatives like RaTG13 unlikely to establish in humans. These results are consistent with modelling studies showing that cross-immunity elicited by human influenza viruses can lower the attack rate of avian-origin influenza viruses with pandemic potential[31]. Further, our results suggest that the deployment of existing COVID-19 vaccines as an intervention measure further reduced the chance of sustained transmission, with more beneficial effects if implemented soon after the first SARS-CoV-X case. In contrast, delays in implementing preventative vaccination will likely erode its effectiveness.

To measure the breadth and potency of cross-protective antibodies, we performed pseudotype-based neutralisation assays using virions carrying the spike proteins of four sarbecoviruses closely related to SARS-CoV-2. Neutralising antibody titers have previously been proposed as correlates of protection against different viral diseases[32], including COVID-19[33], measles[34], and smallpox[35]. Our data suggest that cross-neutralisation between the spike proteins of SARS-CoV-2 and zoonotic sarbecoviruses is associated with their phylogenetic distance. However, this seems at odds with the low neutralisation levels observed against Omicron BA.1.17, which shows low neutralisation despite 96.9% spike identity

with the Wuhan strain (Supplementary Fig. 9). This disparity has also been observed by others[36], and attributed to the unique selection pressures imposed on SARS-CoV-2 to escape prior immunity in humans. Antigenic cartography further supports this, showing that a small number of mutations in the receptor binding domain can cause large antigenic shifts[37].

The highest levels of cross-neutralisation were consistently observed in patients with hybrid immunity, suggesting that vaccine-breakdown infections by immune evasive SARS-CoV-2 variants may have a strong protective effect against SARS-CoV-X infection, and that vaccination should be encouraged even in patients with a history of prior infection. In unvaccinated individuals with a history of infection, the strength of cross-neutralisation was lower than in patients with hybrid immunity and was determined by the SARS-CoV-2 infecting strain. This is consistent with findings that protection conferred by natural infection varies over time and is influenced by the antigenic evolution of SARS-CoV-2, with pre-Omicron infections offering durable immunity, and immunity following Omicron infection waning more rapidly, likely due to increased immune escape[38]. Levels of cross-protection in individuals without vaccine-derived antibodies may

therefore fluctuate according to the antigenic evolution of SARS-CoV-2. Accordingly, genomic and serological surveillance should continue to quantify the extent of cross-neutralisation elicited by new SARS-CoV-2 variants as they arise.

It is conceivable that a zoonotic sarbecovirus could increase its prevalence in animal populations (like H5N1 for influenza). Our results suggest that preemptive vaccination with current COVID-19 vaccines in high-risk regions could help control SARS-CoV-X before human cases emerge. Unlike the variable immunity from repeated SARS-CoV-2 infections, vaccine-induced immunity should be more uniform. Although first-generation COVID-19 vaccines were based on the ancestral Wuhan strain, newer bivalent formulations include both Wuhan and Omicron, potentially offering broader cross-neutralisation. While some studies suggest immunological imprinting may limit this breadth[39,40], others show vaccines targeting conserved sarbecovirus spike regions can improve cross-reactive responses[41,42]. Another important factor to consider is the type of vaccine used. For example, mRNA-based vaccines elicit higher neutralisation titres than adenovirus-vectored vaccines (e.g., ChAdOX), and in some cases higher titres than natural infections[43]. These findings support the development of cross-protective, universal vaccines for sarbecoviruses[44] and other antigenically variable pathogens like influenza[45].

Our study has various limitations. In the absence of more detailed information, we assumed that the in vitro activity of neutralizing antibodies directly correlates to in vivo protection from infection. Quantifying the exact shape of this relationship is challenging, requiring large longitudinal studies of animal or human infections (e.g[46–48].), and existing studies suggest this relationship varies across systems, populations, and study designs[48]. While neutralising antibodies represent a major component of the humoral response, protection in vivo also depends on non-neutralising antibody functions and cellular immunity, which may modify this relationship. In addition, although our data and model account for hybrid immunity, repeated exposures to antigenically distinct SARS-CoV-2 variants further modulate antibody breadth and potency[49,50]. Because detailed infection histories were unavailable for our serum donors, the neutralisation values reported here likely reflect a mixture of singly and multiply exposed individuals and should therefore be viewed as population-weighted averages across varying exposure histories, rather than as discrete infection categories. To account for these uncertainties, we also performed simulations in which cross-immunity and vaccine effectiveness were treated as independent of the neutralisation data. Overall, these simplifications reflect first-order approximations of complex immune processes that provide a framework for understanding emergence dynamics that may be refined in future work.

Our model is conceptual, and more accurate risk assessments will require future models to include the influences of, for example, T cell-mediated cross-immunity[51], spatial and demographic variation in social behaviour and vaccine compliance[52], and evolving immunological landscapes as vaccines and variants change[53]. Our model also assumes a spatially homogeneous, fully mixed population and therefore does not fully capture the strong spatial stochastic effects that dominate during the earliest stages of emergence, when infections are few and often self-extinguishing. As a result, the present framework may modestly overestimate the absolute probability of emergence. Nonetheless, by focusing on the relative effects of key epidemiological and immunological factors, our analysis provides important new information for understanding which parameters most strongly shape emergence risk. We also do not explore here how temporal changes in population immunity, vaccine uptake, or seasonal variations in social contact behaviour or virus transmissibility may influence emergence risk over time, but these represent important extensions for future work. As models become more complex, they must be supported by better methods to quantify virus phenotypes and predict evolutionary trajectories, as effective risk assessments will depend on the combined insights of these elements.

Identifying pathogens that might cause future pandemics is essential for preparedness[27]. The current H5N1 influenza panzootic[54] and the increasing number of human infections by zoonotic H5N1[55] highlight the importance and timeliness of this issue, as does the continuous identification of novel animal coronaviruses able to effectively bind human ACE2 receptors[56]. Both the World Health Organization (WHO) and UK Health Security Agency (UKHSA) have published lists of priority pathogens and diseases with the potential to cause future pandemics, which include zoonotic sarbecoviruses[57,58]. Together, our results indicate that the risk of emergence of novel sarbecoviruses antigenically related to SARS-CoV-2 has decreased due to shifts in population immunity in the post-pandemic era, and that existing endemic viruses may present a strong barrier to new zoonotic threats. Moreover, our study suggests that the rapid and widespread deployment of readily available SARS-CoV-2 vaccines could be an effective strategy to limit the emergence of novel sarbecoviruses that aligns with the 100 Days Mission[59] to respond to future pandemic threats.

## Methods
### Ethics statement
Ethical approval for the collection of serum residual samples from was provided by NHS Greater Glasgow and Clyde Biorepository (application 837).

### Serum samples
Residual biochemistry serum samples were randomly collected from both primary care (general practice) and secondary care (hospital) settings by the NHSGGC Biorepository between March 31, 2020, and September 22, 2021. Informed consent for the use of serum samples was waived by the NHSGGC Biorepository, as the study used anonymised serum samples. Metadata including date of collection, sex, age, and vaccination history were provided, and the natural infection history of each patient was inferred from the date of last positive PCR test. Of the approximately 41,000 samples collected, a representative subsample of 350 sera was selected that maintained the underlying demographic proportions of the full sample by stratified random sampling. Metadata on the complete infection history of each individual was unavailable, and so categories of natural infection history likely include both singly- and multiply-exposed individuals.

### Generation of lentiviral pseudotypes
Gene constructs containing the SARS-CoV (GenBank: AY394995.1), Rs4084 (GenBank: KY417144.1), GX/P1E (GISAID: EPI_ISL_410539), and RaTG13 (GenBank: MN996532.2) spike genes were codon-optimized and synthesized by GenScript Biotech. Each construct was cloned into an expression vector and co-transfected with p8.91-HIV (a gag-pol lentiviral core)[60] and pCSFLW-firefly (which contains a luciferase reporter gene with HIV-1 packaging signal)[61] into HEK293T cells using polyethylenimine. Cells were maintained at 37 °C, 5% $CO_2$, in Dulbecco's modified Eagle's medium (DMEM) supplemented with 10% fetal bovine serum (FBS), 2-mmol/l L-glutamine, 100 μg/ml streptomycin and 100IU/ml penicillin. Supernatants containing HIV (SARS-CoV-X) virions were harvested 48 h after transfection and frozen at −80 °C before use. All plasmids used in this study are available for reuse.

### Neutralization assays
Pseudotype-based neutralization assays were performed as described previously[62,63]. Briefly, stably expressing HEK293-hACE2 (human angiotensin-converting enzyme 2) cells were generated by transfection of HEK293 cells with pSCRPSY-hACE2 and maintained as above. Serum samples were diluted 1:25 in Complete DMEM, and 25 μl of each diluted serum sample was transferred to the wells of a white 96-well plate. An

equal volume (25 μl) of viral pseudotype was added to each well, resulting in a final serum concentration of 1:50. Plates were incubated for 1 h at 37 °C, 5% $CO_2$, after which 50 μl of $4 \times 10^5$ cells/ml of HEK293-hACE2 cells were added to each well. Plates were then incubated for a further 48 h. After incubation, 100 μl of Steadylite Plus was added to each well, and the plate was incubated in the dark for 10 min. Luciferase activity in each well was quantified on a Perkin Elmer Ensight multimode plate reader. The proportion of pseudovirus neutralization was calculated as $n_i = 1 - \frac{L_i}{L_c}$, where $L_i$ is the luciferase activity of sample wells, and $L_c$ the luciferase activity of no-serum control wells. Two technical replicates for each combination of serum and pseudovirus were collected and averaged to provide final measures of neutralization. Where $n_i < 0$, which occurred when the luciferase activity of sample wells exceeded the activity of no-serum control wells, the proportion of neutralization was set to zero.

To estimate overall effects of vaccination and recovery from SARS-CoV-2 infection on neutralization, data from all serum groups were used to fit linear mixed effects models with a fixed effects structure $y_{i,r,v} = \beta_{1,r} + \beta_{2,v} + \beta_{3,r,v} + e$. Here, $y_{i,r,v}$ is the percentage neutralization of serum from individual $i$ with recovery status $r$ and vaccine status $v$. $\beta_{1,r}$, $\beta_{2,v}$, and $\beta_{3,r,v}$ represent the predicted contribution to percentage neutralization of recovery status, vaccination status, and their interaction, respectively, while $e$ represents the model residuals. In this model, random effects of pseudotype virus identity and serum ID were included to account for between-group variability and repeated measures from individual sera. Differences in neutralization between viruses were estimated using a similar modelling approach but with virus identity included as a fixed effect and recovery and vaccine status as random effects. To estimate the overall effect of spike protein similarity on neutralization, data from all serum groups, excluding naïve + unvaccinated individuals, were used to fit a quadratic mixed effects model with a fixed effects structure $y_{i,s} = \beta_s^2 + e$. In this model, $y_{i,s}$ is the percentage neutralization of serum from individual $i$ against spike protein of similarity $s$; $\beta_s^2$ represents the predicted percentage neutralization against a spike protein of similarity $s$. Random effects were included for recovery status, vaccination status, and serum ID. Similarity were calculated as the percentage amino acid identity to the Wuhan-Hu-1 spike protein sequence, taken from a translation alignment (described below).

## Spike protein phylogeny
Spike protein coding sequences of SARS-CoV-2 (Wuhan-Hu-1 strain, GenBank: MN908947.3), SARS-CoV, Rs4084, GX/P1E, and RaTG13 (accessions in the previous section) were translated and aligned in MUSCLE[64] with default settings. Phylogenetic inference was performed with BEAST version 10.5.0[65] using a 1 + 2 & 3 codon partition model with unique substitution rates and base frequencies for the partitions. Each partition was fitted to separate relaxed uncorrelated lognormal molecular clock models[66] using random starting trees, four-category gamma-distributed HKY substitution models, and a birth-death process tree-shaped prior[67]. Four independent BEAST runs were performed, each with 100 million Markov Chain Monte Carlo (MCMC) iterations sampled every 1000 iterations. Runs were combined using LogCombiner with a burn-in of 20% and evaluated for convergence, sampling, and autocorrelation using Tracer version 1.7.2[68]. A maximum clade credibility tree was inferred from the posterior sample and visualized with ggtree[69].

## Epidemiological model structure
An age-structured, stochastic SEIRS model was used to explore the population dynamics of co-circulating SARS-CoV-2 and SARS-CoV-X in the presence of vaccination (Fig. 2A). Within this structure, susceptible (S) individuals may encounter infectious individuals with SARS-CoV-2 (I2) or SARS-CoV-X (IX) and transition to a corresponding exposed compartment (E2 or EX), with the likelihood of S-to-E transition controlled by the transmission probabilities of each virus. Exposed individuals then transition to an infectious state (I2 or IX), with the probability of E-to-I transition inferred from the incubation rate of each virus, and from infectious to recovered (R2 or RX) with a probability inferred from the recovery rates of each virus. Recovered individuals are considered immune to the virus they have recovered from but may transition back to a naïve state over time, with the R-to-S transition probability calculated from the waning rates of infection-acquired immunity.

To allow for re-infection of recovered individuals by the circulating SARS coronavirus, recovered individuals may also, upon encountering an infectious individual, transition to a separate exposed state ($E2_{RX}$ or $EX_{R2}$). These transitions occur with probabilities distinct from those of S-to-E, allowing immunity from one virus to reduce susceptibility to infection with the other. Re-infected individuals then transition through separate infectious compartments ($I2_{RX}$ or $IX_{R2}$) — allowing re-infections to have distinct mortality and transition probabilities to infections of immunologically naïve individuals — before transitioning to the same recovered compartment ($R_{all}$) where individuals are refractory to both viruses. Unique EIR compartments exist for each SARS-CoV-2 variant (Wuhan, Alpha, Delta, Omicron), allowing their infection phenotypes to vary, and additional E and I compartments exist for individuals recovered from a previous SARS-CoV-2 variant that have been re-infected with a subsequent variant. This allows recovered individuals who are immune to one variant (e.g., Wuhan) to be re-infected with a later variant, albeit with some cross-immunity.

Three distinct levels of this base structure were used to represent different vaccination statuses (unvaccinated, "U"; protected by 1 dose, "$V_I$"; and protected by 2 doses, "$V_{II}$"). Individuals in a lower level of vaccine protection may transition to their equivalent compartment in a higher level of vaccine protection with a probability based on the current population vaccination rate, and from a higher level to unvaccinated at a probability based on the rate of decay of SARS-CoV-2 vaccine-derived antibodies. By representing vaccination in this way, a distinction is made between the current protection phenotype of an individual, which increases with vaccination and decreases with waning immunity, and an individual's vaccination history, where the number of doses may only increase with time. This allows an individual with a vaccination history of many doses to have an unprotected phenotype, provided enough time has passed from their last dose for the protection to have waned. Given the timing of vaccinations during and after the COVID-19 pandemic in the model population (Scotland), a negligible number of individuals achieved a protected by 3 doses ("$V_{III}$") phenotype throughout the model run, and so a fourth level representing these individuals was omitted for parsimony (Supplementary Fig. 2A). This model structure allows for individuals with immunity from both natural infection and vaccination (e.g., $R2_{VII}$) to differ phenotypically from those with just natural (e.g., R2) and just vaccine (e.g., $V_{II}$) sources of immunity, capturing some of the diversity in immune protection due to combinations of exposure and vaccine history.

## Host population structure
The population of Scotland was used as the basis for the host model population, for which there is detailed demographic and epidemiological information from before the COVID-19 pandemic to the present. Where information directly describing this population was unavailable, such as in the case of social contact data during and after the pandemic, information from the UK-wide population was used. Each compartment of the model was sub-divided into 16 age groups of 5 years, beginning 0-4 years old and ending 75+ years old, and the starting population sizes of each group were set to the projected population estimates from the National Records of Scotland for 2020.

The birth rates of each age group were set as $\mu_{b,g} = \mu_{f,g} \times \frac{N_{\female,g}}{N_g}$, where $\mu_{b,g}$ is the birth rate, $\mu_{f,g}$ the fertility rate, $N_{\female,g}$ the female population size, and $N_g$ the total population size of age group $g$. All births were added to the 0-4 age group, and increases in age group occurred deterministically at a daily rate of $\frac{1}{365 \times 5}$. Age-stratified mortality and net migration rates were taken from publicly available data (see Supplementary Table 7 for all parameterisation sources). Migrants were assumed to have the same epidemiological properties as the Scotland population at the point of migration. For example, if 5% of the model population were in the I2 compartment, each migrant entering the model at that time point had a probability of 0.05 of being infectious for SARS-CoV-2. These parameters together created a host population that was largely stable over the 8-year duration of the model runs, with size and age progression similar to official population projections (Supplementary Fig. 10).

To represent changes in social contacts over time, questionnaire data from multiple social mixing studies were used to calculate age-specific per-person per-day contact rates for different time periods. For the pre-lockdown period of 01/01/2020 (dd/mm/yyyy) to 23/03/2020, rates were set to those of the 2008 POLYMOD study of the UK population[70], with the study's contact sub-categories ("home", "work", "school", and "other") summed to produce an overall daily rate of contact. For the period of 24/03/2020 to 02/03/2022, UK social contact information was collected continuously by the CoMix study[71]. In this study, the authors split their data into nine timeframes describing different lockdown and easing periods. To provide a finer temporal scale to the epidemiological model, this data was re-analysed to provide weekly estimates of contact rate using the same approach as the original study. Briefly, participant contacts were censored to a daily maximum of 50, and the mean number of contacts inferred from a negative binomial model of structure $m_{i,j} = \beta_{1,i} + \beta_{2,j} + \beta_{3,i,j} + e$. Here, $m_{i,j}$ represents the mean number of contacts of participant age group $i$ to contact age group $j$; $\beta_{1,i}$ is the effect of participant group $i$ on the mean number of contacts; $\beta_{2,j}$ is the effect of contact group $j$ on the mean number of contacts; $\beta_{3,i,j}$ is the interaction effect of participant group $i$ and contact group $j$ on the mean number of contacts; and $e$ is the model residuals.

Symmetrical per-capita contact matrices were calculated from the output of this model using the following equation: $m'_{i,j} = \frac{1}{2N_i}\left(m_{i,j}N_i + m_{j,i}N_j\right)$. Here, $m'_{i,j}$ is the adjusted mean number of contacts, $N_i$ the total number of individuals in age group $i$, and $N_j$ the total number of individuals in age group $j$. A follow-up to the CoMix study was conducted with data collected from 17/11/2022 to 07/12/2022, providing the most up-to-date estimates of UK contact rates available[72]. Data from this study was used to set the model contact rates from 17/11/2022 to the end of the model run (01/01/2028). Weekly contact rates in the inter-study period of 03/03/2022 to 16/11/2022 were inferred by linear interpolation from the final week of data of the original study to the first week of the follow-up study.

**Virus and immune phenotype parameterization**

Parameter values for the infection-related mortality rates and incubation rates (E-to-I) of SARS-CoV and each SARS-CoV-2 variant were obtained from the literature (Supplementary Table 7). Literature estimates of the recovery rate (I-to-R), naturally-acquired immunity waning rates (R2-to-S2), and $R_0$ for SARS-CoV-2 were highly variable and showed little consensus. Instead, these parameters were calibrated by fitting the epidemiological model to prevalence data taken from the UK Coronavirus Infection Survey[30], using an approximate Bayesian computation (ABC) approach as follows.

Log-normal priors were used for the $R_0$ of each variant ($\mu = 1$, $\sigma = 0.5$ on the log scale), corresponding to a median $R_0$ of ~3.0 and a 95% prior credible interval (CI) of ~1–7.2. Logit-normal priors were used for the recovery rates ($\mu = -1$, $\sigma = 0.6$ on the logit scale), corresponding

to a median of 0.26 and a 95% prior CI of 0.10–0.54, and for the immune waning rates ($\mu = -5$, $\sigma = 2$ on the logit scale), corresponding to a median of 0.0067 and a 95% prior CI of 0.0001–0.25. Due to the sequential non-independence of parameters (the values of subsequent variants are highly-dependent on the calibrated estimates of previous variants), calibration was performed separately for each variant in order of emergence. For each variant, 2.5 million randomly sampled parameter sets were run for 10 replicate model iterations, and mean model-estimated SARS-CoV-2 prevalence was calculated for each timepoint present in the Coronavirus Infection Survey. Parameter sets were filtered to remove simulations with a RMSE > 0.01, corresponding to an average deviation of > 1% from the observed SARS-CoV-2 prevalence. For the remaining parameter sets, Epanechnikov kernel weights were applied to scaled RMSE values to construct a smooth approximation of the posterior distribution. To propagate uncertainty in these parameter values forward, all subsequent models included in this study sampled unique values from each approximate posterior for each independent model iteration. For SARS-CoV, literature estimates of $R_0$ were available, but no estimates were available for the recovery rate. Additionally, no information on the infection phenotypes of the prospective zoonotic SARS coronaviruses in humans exists. In the absence of this information, all missing phenotypes were assumed to equal those of the most closely related SARS coronavirus with a known phenotype. For example, the $R_0$ of Rs4084 was set equal to SARS-CoV, while those of RaTG13 and GX/P1E were set equal to the SARS-CoV-2 Wuhan variant.

Transmission rates were calculated from the following $R_0$ equation, derived from the SEIRS model structure: $R_{0,v} = \frac{\varepsilon_v \beta_v m'}{(\varepsilon_v + \lambda)(\gamma_v + \delta_v + \lambda)}$. Here, $\beta_v$ is the transmission rate; $R_{0,v}$ the basic reproductive number; $\varepsilon_v$ the incubation rate; $\gamma_v$ the recovery rate; and $\delta_v$ the infection-related mortality rate of virus $v$. $\lambda$ is the population crude death rate and $m'$ the mean per-day per-capita contact rate. The transmission rates and infection-related mortality rates above describe the phenotypes of infection in naïve and unvaccinated individuals. Estimates of the proportion with which infection-related mortality is reduced in recovered and vaccinated individuals were taken from the literature. The proportion with which transmission rates are reduced in recovered and vaccinated individuals for SARS-CoV-X infections is assumed to correspond linearly $(1 - x)$ to the pseudotype neutralization data in Fig. 1, and for SARS-CoV-2 re-infections from[62] (Supplementary Fig. 9). The exact relationship between neutralization titres and protection from sarbecovirus infection is unknown, with both linear and non-linear relationships reported in the literature[5].

The waning rates of immunity in recovered individuals (R-to-S) and vaccinated individuals ($V_I$-to-S and $V_{II}$-to-S) were calculated separately from the half-lives of infection and vaccine derived antibodies against SARS-CoV-2 with the following exponential decay equation: $\mu_{w,v} = \frac{-\ln(0.5)}{t_{\frac{1}{2},i}}$. Here, $\mu_{w,v}$ represents the waning rate, and $t_{\frac{1}{2},v}$ the antibody half-life of immunity of type $i$ (natural or vaccine).

**Age-stratified phenotypes**

There is strong evidence in the literature that immune protection and infection-related mortality for coronavirus infections vary across age groups. To represent this variation, non-linear least squares models, fitted using the nls function from the R core stats package[73], were used to infer how the relationship between phenotype and age changes with the population parameter value. For infection-related mortality, age-stratified data on SARS-CoV-2 and MERS-CoV were used to fit a four-parameter logistic function (Supplementary Fig. 11) as $\delta_{v,a} = L_2 + \beta_1 \delta_v + \frac{\beta_2 \delta_v + L_1 - L_2}{1 + e^{-(k + \beta_3 \delta_v)(a - a_0 + \beta_4 \delta_v)}}$. Here, $\delta_{v,a}$ is the infection-related mortality rate of virus $v$ for hosts of age $a$; $k$ is the slope, $L_1$ the upper plateau, and $L_2$ the lower plateau of the logistic function; and $a_0$ is the

inflection point. $\beta_1\delta_v$, $\beta_2\delta_v$, $\beta_3\delta_v$, and $\beta_4\delta_v$ describe how the population infection-related mortality rate affects the lower plateau, upper plateau, slope, and midpoint of the logistic function, respectively.

Evidence for age-stratified effects of vaccine-efficacy are also prevalent in the literature, although effect sizes are typically minor[74–79]. The effectiveness of the second dose of the COVID-19 vaccine, administered at different intervals from the first dose, was used to provide a dataset where the population-level estimate of immune protection varied, and the age-stratified estimates of immune protection were known (Supplementary Fig. 12). This data was used to fit a model with a formula composed of two three-parameter logistic functions (one describing the increase in immune effectiveness from birth to middle age, the other describing the decline in effectiveness from middle age to old age), and an intercept influenced by the population-level protection: $p_{i,a,v} = L_3 + \beta_1 p_{i,v} + \frac{L_1}{1+e^{-k_1(a-a_{1,0})}} + \frac{L_2}{1+e^{-k_2(a-a_{2,0})}}$. Here, $p_{i,a}$ is the level of protection conferred by immunity source $i$ against virus $v$ on age group $a$. $L_1$, $k_1$, and $a_{1,o}$ are the upper plateau, slope, and infection point of the increase in immune effectiveness with age, while $L_2$, $k_2$, and $a_{2,o}$ are the equivalents for the decrease in immune effectiveness with age. $\beta_1 p_{i,v}$ describes the effect of population-level immune protection on the function's intercept ($L_3$). Little consensus exists for an effect of host age on the incubation or recovery rates of SARS coronavirus infections[80–84], and so these parameters are assumed to be equal across age groups.

## Vaccination rates

For the initial COVID-19 vaccination program (08/12/2020 to 11/09/2022), daily vaccination rates for each age group receiving the first ($u_{D_1}$), second ($u_{D_2}$), and third ($u_{D_3}$) vaccine doses were calculated from publicly available data[85]. As vaccination exists in the epidemiological model as a transient phenotype with three levels (U, $V_I$, $V_{II}$), two distinct rates exist: the transition rate from naïve to protected with 1 dose ($u_{U \to V_I}$) and the transition rate from protected with 1 dose to protected with 2 doses ($u_{V_I \to V_{II}}$). As entry into the $V_{II}$ level is impossible while only 1 dose of the vaccine has been administered, this rate was set as $u_{V_I \to V_{II}} = u_{D_2} + u_{D_3}$. Transitioning from a naïve to vaccinated phenotype can be produced at any point in the vaccination program and by any dose in a patient's vaccination history, and so this rate was set as $u_{U \to V_I} = u_{D_1} + u_{D_2} + u_{D_3}$. Vaccination rates by age group during the booster programs in Winter 2022, Spring 2023, Winter 2023, and Spring 2024 were used to extend $u_{D_3}$ to 14/07/2024[86]. Spring and Winter booster programs were then assumed to recur annually at rates identical to Winter 2023 and Spring 2024 for the remainder of the model.

The rate of administration of the first COVID-19 vaccine dose over time in Scotland is well approximated by a non-normalised Gaussian curve (Supplementary Fig. 13), and so a similar curve with a mean (μ) of 30 days and standard deviation (σ) of 10 days was used to define a theoretical preventative campaign against SARS-CoV-X emergence lasting 60 days: $r(t) = Ae^{-\frac{(t-\mu)^2}{2\sigma^2}}$. By adjusting the scaling factor (A), the vaccination rates described by this curve result in different cumulative vaccine uptake levels over the course of the campaign. Solutions for A that produce specific values of vaccine uptake were calculated using root-finding methods with the uniroot function in R[73].

## Stochasticity

Mean rates for all model parameters (excluding the aging rate) were converted to probabilities using a Euler scheme with binomial increments and exponential rates[87] such that the number of individuals $N$ transitioning from compartment $A$ to $B$ at time $t$ is given as $N_{A \to B,t} = Binomial(N_{A,t-1}, 1 - e^{-\mu_{A \to B,t}})$. Here, $Binomial(n, p)$ represents the binomial distribution, $n$ is the number of individuals available to a given transition, and $p$ is the probability of transition calculated from the mean rate $u$.

## Model run conditions

The model was implemented and run using the Odin[88] and Odin Dust[89] R packages, with random starting seeds, a 1-day time step, and a duration of 8 years spanning 1st January 2020 to 1st January 2028. To introduce the SARS-CoV-2 Wuhan variant, five exposed individuals (E2_Wuhan) were added to the model on 23rd February 2020. This date was inferred from the description of the first confirmed positive case of SARS-CoV-2 infection in Scotland[90], which was reported on 2nd March 2020 in an individual presenting 3 days after symptom onset (-3 days) and after a ~ 5-day incubation period (-5 days). Five individuals were chosen as this ensured SARS-CoV-2 reached endemicity in all model runs conducted in this study. Introduction of new SARS-CoV-2 variants to the model occurred deterministically by taking the total number of SARS-CoV-2 infectious individuals, regardless of variant, and multiplying by the prevalence of each variant as reported for the Scotland population over time. These values were then used in conjunction with contact probabilities to calculate the number of individuals exposed to each variant as the model progressed. A single 30-year-old SARS-CoV-X individual (EX) was introduced to the model on 23rd February 2024, four years after the initial SARS-CoV-2 exposure.

## Sensitivity analysis

To provide a sufficient sampling of parameter space to allow for a Sobol sensitivity analysis on nine parameters—the $R_0$ of SARS-CoV-2 and SARS-CoV-X; the levels of natural and vaccine-acquired cross-immune protection against SARS-CoV-X; the timing and uptake of the preventative vaccination program; and the waning rates of vaccine-acquired immunity, naturally-acquired SARS-CoV-2 immunity, and naturally-acquired SARS-CoV-X immunity—a machine learning emulator was trained on the output of the epidemiological model as follows. A Sobol sample (n = 500,000) of nine-dimensional parameter space was produced using uniform distributions of each parameter (Table 1). Each parameter set in the Sobol sample scheme was run for a small number of iterations (500), providing a dense but high-Monte-Carlo-noise exploration of parameter space, an approach that has been shown to efficiently provide training data for model emulation[91]. The probability of SARS-CoV-X emergence (emergences/trials) was then used to train a gradient-boosted regression trees (GBRT) model using the R package XGBoost[92], configured with a maximum tree depth of 9 (equal to the number of explanatory variables), a learning rate of 0.01, a subsampling fraction of 0.7, and all available predictors included at each tree split. Ten-fold cross-validation was used within XGBoost to identify the optimal number of boosting iterations to avoid overfitting the training data. An additional 5000 parameter sets were produced using a Latin Hypercube sample of parameter space with the same uniform distributions on each parameter. Each of these parameter sets were run for 5000 model iterations to produce a low-density, low-Monte-Carlo-noise test dataset. When evaluated on the independent test dataset, the trained emulator closely reproduced the epidemiological model outputs, achieving an $R^2 > 0.997$ with errors consistent across predicted values (Supplementary Fig. 14). Sensitivity analysis was then conducted using variance-based Sobol indices implemented in the R package Sensitivity[93]. A Saltelli paired-sample design was generated from two $200,000 \times 9$ Sobol sampling matrices, for a total of 2.2 million parameter sets. Predicted probabilities of SARS-CoV-X emergence for each parameter set were taken from the GBRT emulator and used to calculate first-order and total-order Sobol indices, with 1000 bootstrap replicates to estimate confidence intervals.

**Reporting summary**

Further information on research design is available in the Nature Portfolio Reporting Summary linked to this article.

## Data availability

All data used in this study can be found on GitHub (https://github.com/ryanmimrie/Publications-2026-SARS-CoV-X-Emergence) and FigShare (https://doi.org/10.6084/m9.figshare.28566677.v1).

## Code availability

All models, code, and statistical analyses used in this study can be found on GitHub (https://github.com/ryanmimrie/Publications-2026-SARS-CoV-X-Emergence) and FigShare (https://doi.org/10.6084/m9.figshare.28566677.v1).

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

## Acknowledgements

This study was funded by the following sources: Medical Research Council MC_UU_0034/3 (PRM), MC_UU_00034/6 (BJW) & MR/Y002814/1 Biotechnology and Biological Sciences Research Council BB/V004697/1 (PRM, MV). We thank the NHS Greater Glasgow and Clyde Biorepository for providing serum samples, and the editor and anonymous reviewers for their constructive comments.

## Author contributions

Conceptualization: M.V., B.J.W., P.R.M. Methodology: R.I., M.V., B.J.W., L.M., M.B., P.R.M. Investigation: R.I., L.A.B., S.R., M.M., J.A.R.A., L.M., N.L. Visualization: R.I., L.A.B., M.M., J.A.R.A. Funding acquisition: P.R.M., B.J.W., M.V. Project administration: P.R.M., B.J.W. Supervision: M.V., B.J.W., P.R.M. Writing—original draft: R.I., L.A.B., M.V., B.J.W., P.R.M. Writing—review & editing: R.I., L.A.B., S.R., M.M., J.A.R.A., L.M., N.L., A.P., M.B., M.V., B.J.W., P.R.M.

## Competing interests

The authors declare no competing interests.

## Additional information

[1]MRC-University of Glasgow Centre for Virus Research, Glasgow, UK. [2]School of Biodiversity, One Health and Veterinary Medicine, University of Glasgow, Glasgow, UK. [3]Department of Laboratory Medicine, Yale School of Medicine, New Haven, USA. [4]Odum School of Ecology, University of Georgia, Athens, GA, USA. [5]MRC Centre for Global Infectious Disease Analysis, Abdul Latif Jameel Institute for Disease and Emergency Analytics (J-IDEA), School of Public Health, Imperial College London, London, UK. [6]Centre for Mathematical Modelling of Infectious Diseases, London School of Hygiene and Tropical Medicine, London, UK. [7]These authors contributed equally: Ryan M. Imrie, Laura A. Bissett. ✉e-mail: Mafalda.Viana@Glasgow.ac.uk; Brian.Willett@Glasgow.ac.uk; Pablo.Murcia@Glasgow.ac.uk

