## [Transparent Peer Review file · Nature Communications]

Post-pandemic changes in population immunity have reduced the likelihood of emergence of zoonotic coronaviruses

Corresponding Author: Professor Pablo Murcia

Version 0:

Reviewer comments:

Reviewer #1

(Remarks to the Author)

Summary

The authors use a combination of neutralising antibody data, sequence data, phylogenetic tree inference and a detailed transmission model to estimate the probability of the emergence of novel zoonotic coronaviruses, given the levels of immunity to existing coronaviruses and the estimated cross-reactivity and cross-protection conferred by such cross-reactivity. The aims of the study are certainly worthwhile, given how valuable knowledge of the likelihood of a novel zoonosis would be, to prepare well for the resulting wave(s) of infections.

While the questions are important and much of the work is impressive, the ability to accurately estimate the emergence of a novel pathogen is extremely difficult. There are numerous, disparate datasets and analysis techniques used in conjunction with each other throughout this study. A consequence of this approach is that each dataset, analysis technique and set of results used to parameterise the final transmission model has its own set of limitations and assumptions. Furthermore, some parts of the analysis involve assumptions or limitations that are strong enough on their own to put in doubt the validity of the conclusions of the study.

Clearly, the authors have put a great deal of work into the various analysis streams: the phylogenetic tree estimation, the cross-reactivity used to parameterise cross-protection in the transmission model, the incredibly detailed and long transmission model itself, etc. However, we do think that the text and supplement do not describe any one part of the study with enough rigour or detail. We would struggle to replicate or even understand any one of these streams from the two texts. The codebase allows the user to see that the code is over 7000 lines long, which implies that the transmission model developed and used is incredibly detailed. However, the details on the fitting itself are scarce. They just briefly outline how the authors performed a grid search and fit to the ONS data. Grid search is close to the simplest possible method that could be used to fit a transmission model to data — so simple in fact we think most critics would argue it does not count as fitting the model. It just seems bizarre to develop and parameterise such a complex transmission model, but not to validate it, report its ability to reconstruct parts of the pandemic that have already occurred or to implement a fitting approach which is somewhat statistically robust. The fact that no uncertainty is included in the final estimates is also dubious in a similar fashion. Overall, the message, both of the focus of the paper, but also the detail, care and confidence in the results is confusing throughout the study. Clearly, it is a difficult and ambitious task. However, we think it would be arguable that performing any one of the subtasks well enough to be publishable — the phylogenetic tree estimation, cross-reactivity and cross-protection estimates, a detailed enough transmission model — on its own. Combining them all into one large task, where either each subtask was performed relatively badly, or just reported in a confusing manner, does the entire study and each component a disservice. As it stands, we struggle to see how this study as it is currently structured could be of publishable standard. We believe that including much more detail on the level of rigour and statistical robustness for each step of the analysis, significantly improving the quality of the figures and somehow validating the quality of the fits of the transmission model, either against additional data or just by showing that the trajectory, with uncertainty, matches well with the ONS infection survey data, are all required before the study would be publishable. Changing the focus would also be an option, including much more detail about parts of the analysis and convincing the reader that the analysis was robust.

Specific comments

1. The discrepancy between the modelled estimate and the ONS infection survey data on the prevalence of SARS-CoV-2 (Figure 2A) is a clear example of confusion and potential rigour. The supplement outlines how the ONS infection survey data was directly used to perform a grid search approach to fitting the model. However, even though the model was fit to this data, the estimates from the model disagreed with the data for large portions of the pandemic. The ONS infection survey was one of, if not the most valuable and reliable data source during the pandemic in the U.K., as it provided close to unbiased estimates of the prevalence. As such, modelled estimates that fit to this data, but do not agree with the data for long portions of the pandemic, should be treated with caution.
2. Furthermore, the modelled prevalence estimates, both the retrospective and prospective portions, do not appear to have any uncertainty.
3. Figures 2D, 3 and 4 are all very difficult to interpret properly. The colour legends do not go between 0 and 1, they all stop prematurely, which looks very strange. We suppose the data just needs to be normalised, but this alone makes them all difficult to interpret. In every case, it seems as though too much is being varied and reported in one go for it to be digestible. Furthermore, in many cases each panel is almost entirely one single colour, which defeats the point somewhat of a heatmap. We believe a fair bit more attention to these figures is required. Perhaps a total rethink of how these results is presented, as their current form is almost uninterpretable.
4. The assumptions behind parameterising the transmission model using the neutralisation data are large and underdiscussed. The neutralisation data itself is nice, and it is certainly possible to attempt to parameterise estimates of cross-reactivity, and to a certain extent, cross-protection, from such data. However, the current approach is simplistic and not varied or tested. More attention needs to be given to how the neutralisation data ultimately informs the parameterisation of the transmission model, as it is one of the key novel steps of this study.
5. To be honest, we believe the focus of the study needs to be sharpened substantially. Is the focus merging the various and somewhat disparate techniques to arrive at these types of estimates? Or is it, as it seems as though it is currently written, the reporting of what are intended to be accurate estimates of the probability of emergence of a novel coronavirus? Clearly, the latter is the current focus, and the former is only briefly discussed. However, given how hard it is to accurately estimate such a quantity, we believe the study would be a more valuable contribution if the focus shifted. For example, breaking up and reporting the analysis streams would be one option.

Dr Tim Russell and Dr Charlotte Chaloner

(Remarks on code availability)

The code is available on a public Github repository, with a README and enough structure to follow what was done. The repo is well structured and simple enough to allow a user to review what precisely was done. No code or data for the neutralisation or the phylogenetic tree inference were provided, just the transmission model code and parameterisation.

Reviewer #2

(Remarks to the Author)

This manuscript presents a timely investigation into how post-pandemic population immunity—derived from both SARS-CoV-2 infection and vaccination—affects the emergence potential of zoonotic sarbecoviruses. A strength of the manuscript is how the authors combine *in vitro* neutralization assays (quantifying cross-immunity of SARS CoV-2 infection and vaccine immunity) with a thoughtfully constructed stochastic SEIRS model. With this they demonstrate that current levels of cross-immunity significantly reduce the likelihood of novel sarbecoviruses establishing sustained human transmission. Particularly commendable is the integration of empirical immunological data with dynamic modeling. This gives the study relevance to real-world pandemic preparedness. The exploration of both beneficial and potentially adverse effects of vaccination strategies adds nuance and depth to the findings. Overall, this study makes a valuable contribution to our understanding of SARS-CoV-X emergence in the post-COVID era and offers actionable insights for future vaccine development and surveillance strategies. I do have a few comments and concerns, as laid out below.

Methods

Page 7 – It is unclear why the model incorporates age-specific vaccine effectiveness. While it is well-established that COVID-19 outcomes vary significantly by age, the evidence supporting meaningful variation in vaccine effectiveness across age groups is less familiar. Clarifying the rationale for this added complexity would help justify the model's structure.

Figure S2 – The reported one-dose COVID-19 vaccine coverage appears surprisingly low, never exceeding approximately 25%. This seems inconsistent with known vaccination rates in Scotland. Additionally, the model suggests that by 2022, nearly all individuals are categorized as "unvaccinated," which appears to result from a high rate of vaccine waning that outpaces new vaccinations. This assumption may substantially underestimate actual population immunity and warrants further explanation or validation.

Results

The results shown in Figure 4, where natural infection provides high cross-immunity but vaccines do not, appear more extreme and less plausible than those in Figure 3, where vaccine and infection-derived immunity are similar. It would be helpful to see intermediate scenarios—for example, where vaccine-derived immunity is half as effective as infection-derived immunity—to assess whether the conclusions hold under more moderate assumptions.

While the authors explore a broad parameter space in their simulations, all scenarios focus on SARS-CoV-X emergence in 2024. Although the timing of vaccination relative to emergence is examined, it would strengthen the analysis to include simulations that assess risk over time. This is particularly relevant given the likely decline in population immunity as SARS-CoV-2 incidence remains low and vaccine uptake plateaus.

Discussion

The authors acknowledge that their model assumes neutralizing antibody levels are a direct correlate of protection. However, this is a simplification, as immunity to SARS-CoV-2 involves multiple components, and protection against infection (which is most relevant for emergence risk) may differ from protection against disease. A more nuanced discussion of this limitation—and its implications for interpreting the model's predictions—would be valuable.

(Remarks on code availability)

Reviewer #3

(Remarks to the Author)

See the attached reviewer comments files.

(Remarks on code availability)

I have not run the code but it's shared on Github which is best practice and gold standard for open-source code. I've reviewed it and looks good and applaud the authors for sharing transparently in this way.

Reviewer #4

(Remarks to the Author)

(Remarks on code availability)

Version 1:

Reviewer comments:

Reviewer #1

(Remarks to the Author)

The authors have responded to the reviewers' comments, mostly by adjusting or adding text to the manuscript, but also by honing and finalising the figures. Figure 2 is now much improved in terms of readability, it is now a nice figure overall. The heatmaps in Figures 4 and 5 include many panels, most of which are homogeneously coloured, which feels a little in vain. However, the overall pattern is easier to interpret now. Overall, the changes have substantially improved the manuscript.

We believe that the value of the contributions from this work are a little hard to glean. This is partly due to the complexity involved in the aims of the work - estimating the likelihood of the emergence of an established and transmitting novel coronavirus. Showing that widespread protection against SARS-CoV-2, via infections and vaccinations, would likely provide some cross-protection against related sarbecoviruses, is useful to a certain extent. However, the following findings within the paper are a direct consequence of the assumptions within the study. Specifically, assuming a (linear) relationship between neutralisation of related viruses and protection.

However, it is still a nice contribution to see precisely, given the detailed parameterisation of this transmission model, what the quantitative outcomes of such an assumption would be. We believe that this study, in its current format, provide a nice framework for future work. We also believe that the limitations and assumptions are much more clearly laid out in the revised manuscript. As such, we think that it would make a nice contribution to Nature Communications.

(Remarks on code availability)

The codebase has significantly improved in the revised version of the manuscript. It is now clearly structured, includes a README and has scripts and functions to cover all of the analyses performed in the study.

Reviewer #2

(Remarks to the Author)

This is a complex paper that integrates a lot of disparate data, employing a range of methods. The other reviewers and I pointed out several areas for improvement and - even more so - acknowledgement of limitations. I feel that the limitations are now adequately outlined in the paper. My other concerns have also been addressed.

(Remarks on code availability)

Reviewer #3

(Remarks to the Author)

Thanks for materially and meaningfully addressing the comments I raised - this is a very strong manuscript and I'm excited to see it out in the wider world. Congratulations!

(Remarks on code availability)

Reviewer #4

(Remarks to the Author)

(Remarks on code availability)

Reviewer Comments

Reviewer #1 (Remarks to the Author):

The authors use a combination of neutralising antibody data, sequence data, phylogenetic tree inference and a detailed transmission model to estimate the probability of the emergence of novel zoonotic coronaviruses, given the levels of immunity to existing coronaviruses and the estimated cross-reactivity and cross-protection conferred by such cross-reactivity. The aims of the study are certainly worthwhile, given how valuable knowledge of the likelihood of a novel zoonosis would be, to prepare well for the resulting wave(s) of infections.

While the questions are important and much of the work is impressive, the ability to accurately estimate the emergence of a novel pathogen is extremely difficult. There are numerous, disparate datasets and analysis techniques used in conjunction with each other throughout this study. A consequence of this approach is that each dataset, analysis technique and set of results used to parameterise the final transmission model has its own set of limitations and assumptions. Furthermore, some parts of the analysis involve assumptions or limitations that are strong enough on their own to put in doubt the validity of the conclusions of the study.

Clearly, the authors have put a great deal of work into the various analysis streams: the phylogenetic tree estimation, the cross-reactivity used to parameterise cross-protection in the transmission model, the incredibly detailed and long transmission model itself, etc. However, we do think that the text and supplement do not describe any one part of the study with enough rigour or detail. We would struggle to replicate or even understand any one of these streams from the two texts. The codebase allows the user to see that the `odin` code is over 7000 lines long, which implies that the transmission model developed and used is incredibly detailed. However, the details on the fitting itself are scarce. They just briefly outline how the authors performed a grid search and fit to the ONS data. Grid search is close to the simplest possible method that could be used to fit a transmission model to data — so simple in fact we think most critics would argue it does not count as fitting the model. It just seems bizarre to develop and parameterise such a complex transmission model, but not to validate it, report its ability to reconstruct parts of the pandemic that have already occurred or to implement a fitting approach which is somewhat statistically robust. The fact that no uncertainty is included in the final estimates is also dubious in a similar fashion. Overall, the message, both of the focus of the paper, but also the detail, care and confidence in the results is confusing throughout the study. Clearly, it is a difficult and ambitious task. However, we think it would be arguable that performing any one of the subtasks well enough to be publishable — the phylogenetic tree estimation, cross-reactivity and cross-protection estimates, a detailed enough transmission model — on its own. Combining them all into one large task, where either each subtask was performed relatively badly, or just reported in a confusing manner, does the entire study and each component a disservice. As it stands, we struggle to see how this study as it is currently structured could be of publishable standard. We believe that including much more detail on the level of rigour and statistical robustness for each step of the analysis, significantly improving the quality of the figures and somehow validating the quality of the fits of the transmission model, either against additional data or just by showing that the trajectory, with uncertainty, matches well with the ONS infection survey data, are all required before the study would be publishable. Changing the focus would also be an option, including much more detail about parts of the analysis and convincing the reader that the analysis was robust.

Comments

1. The discrepancy between the modelled estimate and the ONS infection survey data on the prevalence of SARS-CoV-2 (Figure 2A) is a clear example of confusion and potential rigour. The supplement outlines how the ONS infection survey data was directly used to perform a grid search approach to fitting the model. However, even though the model was fit to this data, the estimates from the model disagreed with the data for large portions of the pandemic. The ONS infection survey was one of, if not the most valuable and reliable data source during the pandemic in the U.K., as it provided close to

unbiased estimates of the prevalence. As such, modelled estimates that fit to this data, but do not agree with the data for long portions of the pandemic, should be treated with caution.

We thank the reviewers for their comments. We would like to respectfully clarify that the behaviour they have observed in the example model output presented in Figure 2A does not reflect confusion or a lack of rigour in our approach. The purpose of our model is to explore conceptually the consequences of cross-immunity, both vaccine and naturally derived, on the emergence potential of a novel human virus, and we ground this research question in the familiar context of the recent SARS-CoV-2 pandemic. This context provides us with several advantages, including access to a large biobank of relevant human sera for quantification, a wealth of published studies and public datasets on SARS-CoV-2 and related sarbecoviruses, and detailed vaccination and social contact data with which to parameterise our model.

While we have attempted, wherever practical, to make our model an accurate simplification of the epidemiological conditions in Scotland from March 2020 to the present day, it was not our intention to create a model capable of recapitulating all aspects of the ONS infection survey dataset. We currently have represented in our model the SARS-CoV-2 Wuhan, Alpha, Delta, and Omicron variants. As a representation of immune escape, each successive variant is provided with a means to re-infect individuals recovered from infection with a previous variant. This immune escape mechanism allows the model to reproduce sharp changes in prevalence from the ONS data, such as the initial epidemic peak of Omicron, which would otherwise be prevented by the number of Delta recovered individuals in the population.

Four additional sharp peaks are apparent in the ONS data after this initial Omicron peak, which our model does not reproduce, but instead approaches an equilibrium in line with the mean prevalence across this final portion of the ONS data. Theoretically, had we included four additional successive Omicron variants as separate viruses in our model structure, these peaks could be reproduced by immune escape. However, this would double the complexity of our representation of SARS-CoV-2 in the model, decreasing the accessibility of the model code (already noted by the reviewers to be complex), while ultimately achieving the same equilibrium prevalence by the point of SARS-CoV-X exposure. Additionally, we lack the representation of separate Omicron variants in our biobank samples or pseudotype neutralisation data, and so the immune cross-reactivity of each Omicron variant would have to be assumed to be identical. In this sense, we do not think that expanding the model to more closely follow the end portion of the ONS data will provide any additional insights or changes to the results of this study.

To clarify these points, the following additions to the manuscript results have been made:

(Lines 171-176): “The model closely tracked real-world SARS-CoV-2 prevalence until the initial Omicron wave (Figure 2C), after which serological, vaccination, and contact data for Scotland became more limited. Without this information, the additional complexity of modelling identical successive immune-escape Omicron lineages was not justified, and SARS-CoV-2 prevalence instead stabilises at 2.2%, corresponding to the mean prevalence across the Omicron peaks, avoiding unsupported assumptions about short-term variant-specific dynamics.”

2. Furthermore, the modelled prevalence estimates, both the retrospective and prospective portions, do not appear to have any uncertainty.

We have added 95% confidence intervals to any text references to prevalence, and provide a visual representation of this uncertainty in the updated Figure 2A (now Figure 2B). In addition, Supplementary Table 8 now reports the prior and approximate posterior distributions of the calibrated SARS-CoV-2 parameters by variant, providing a description of uncertainty in our calibrated parameters.

The following has now been added to the results:

(Lines 167-171): “To provide estimates for R_0 , the recovery rate, and immune waning rates for each SARS-CoV-2 variant, which vary considerably across the available literature, we calibrated our model using an approximate-Bayesian computation (ABC) process fitted to prevalence data from the Office for National Statistics (ONS) Infection Survey Dataset³⁰ (see Supplementary Table 8 for prior and approximate posterior summaries).”

3. Figures 2D, 3 and 4 are all very difficult to interpret properly. The colour legends do not go between 0 and 1, they all stop prematurely, which looks very strange. We suppose the data just needs to be normalised, but this alone makes them all difficult to interpret. In every case, it seems as though too much is being varied and reported in one go for it to be digestible. Furthermore, in many cases each panel is almost entirely one single colour, which defeats the point somewhat of a heatmap. We believe a fair bit more attention to these figures is required. Perhaps a total rethink of how these results is presented, as their current form is almost uninterpretable.

We thank the reviewer for their comments regarding Figures 2D, 3, and 4 and we have revised the heatmaps substantially. We have used heatmaps in a facet grid as they are useful for intuitively visualising dimensionally complex datasets (e.g., the 12-parameter-space projections in 10.1098/rsos.210506). Ideally, we would keep the same colour range throughout all the heatmaps, however the range of emergence probabilities of our four sarbecoviruses in Figure 2D (0 to 0.111) was considerably smaller than for our theoretical virus explorations in Figures 3 and 4 (0 to 0.75) and keeping the same colour range throughout would render Figure 2D unintelligible. We are hesitant to apply a form of normalisation to our data, as this obscures the ease of interpretability of the likelihood of emergence (a probability between 0 and 1). Instead, we chose to use a smaller colour range for Figure 2D, and then use a consistent range for the remaining Figures in both the main body and supplementary.

In their current format, each heatmap displays two aspects of our model output to the reader: 1) the estimated background emergence probability of SARS-CoV-X, and 2) the estimated probability of SARS-CoV-X emerging under different vaccination regimens. We believe having both aspects in the same heatmap may be contributing to the confusion, and it would be beneficial to interpretability to separate these into 1) a small grid of the background emergence probabilities, and 2) a heatmap of the *change* in emergence probability due to vaccination. This will be especially helpful where effects of vaccination relative to background probabilities are subtle, such as in several panels of Figure 4. We have now updated Figures 2D (now Figure 3), Figure 3 (now Figure 4), Figure 4 (now Figure 5), and Supplementary Figure 5 to this new format.

For readers who prefer an alternative representation of this data, Supplementary Figures 2, 4, and 6 continue to provide complementary visualisations of the same results.

4. The assumptions behind parameterising the transmission model using the neutralisation data are large and underdiscussed. The neutralisation data itself is nice, and it is certainly possible to attempt to parameterise estimates of cross-reactivity, and to a certain extent, cross-protection, from such data. However, the current approach is simplistic and not varied or tested. More attention needs to be given to how the neutralisation data ultimately informs the parameterisation of the transmission model, as it is one of the key novel steps of this study.

We thank the reviewer for highlighting this point. We agree that the link between *in vitro* neutralisation and *in vivo* cross-protection is a key assumption in our parameterisation. However, as discussed in recent reviews (e.g. 10.1038/s41577-023-00858-w), this relationship is exceptionally difficult to quantify in a generalisable way. While the exact form of the relationship varies by virus and context and may be linear (as we assume here) or nonlinear, a consistent observation across many studies is that higher neutralisation titres are associated with higher levels of protection. In this sense, the

direction of the effects reported in this conceptual study is robust to our assumptions on the shape of the relationship between neutralisation and cross-protection. A detailed exploration of the consequences of different assumed relationships between neutralisation and cross-protection is outside the scope of the present study. To highlight this, and to more thoroughly discuss the implications of this assumption on our current study, we have expanded the discussion as follows:

Lines 308-322: “Our study has various limitations. In the absence of more detailed information, we assumed that the *in vitro* activity of neutralizing antibodies directly correlates to *in vivo* protection from infection. Quantifying the exact shape of this relationship is challenging, requiring large longitudinal studies of animal or human infections (e.g. ⁴⁶⁻⁴⁸), and existing studies suggest this relationship varies across systems, populations, and study designs⁵. While neutralising antibodies represent a major component of the humoral response, protection *in vivo* also depends on non-neutralising antibody functions and cellular immunity, which may modify this relationship. In addition, although our data and model account for hybrid immunity, repeated exposures to antigenically distinct SARS-CoV-2 variants further modulate antibody breadth and potency.^{49,50} Because detailed infection histories were unavailable for our serum donors, the neutralisation values reported here likely reflect a mixture of a mixture of individuals that have experienced single or multiple virus exposure events, and should therefore be viewed as population-weighted averages across varying exposure histories, rather than as discrete infection categories. To account for these uncertainties, we also performed theoretical explorations in which cross-immunity and vaccine effectiveness were treated as independent of the neutralisation data. Overall, these simplifications reflect first-order approximations of complex immune processes that provide a framework for understanding emergence dynamics that may be refined in future work”

5. To be honest, we believe the focus of the study needs to be sharpened substantially. Is the focus merging the various and somewhat disparate techniques to arrive at these types of estimates? Or is it, as it seems as though it is currently written, the reporting of what are intended to be accurate estimates of the probability of emergence of a novel coronavirus? Clearly, the latter is the current focus, and the former is only briefly discussed. However, given how hard it is to accurately estimate such a quantity, we believe the study would be a more valuable contribution if the focus shifted. For example, breaking up and reporting the analysis streams would be one option.

Our study is a conceptual exploration of the consequences of cross-immunity on virus emergence, grounded in the context of the recent SARS-CoV-2 pandemic. We have made additions to the abstract, introduction, and discussion to try to make this clear:

(Lines 38-40) “To this end, we combined empirical cross-neutralisation data with mathematical modelling and identified the main immunological and epidemiological factors likely to shape sarbecovirus emergence.”

(Lines 120-122) “These simulations provide a conceptual exploration of how population immunity, vaccination, and viral transmissibility interact to influence emergence.”

(Lines 321-322) “Overall, these simplifications reflect first-order approximations of complex immune processes that provide a framework for understanding emergence dynamics that may be refined in future work.”

Dr Tim Russell and Dr Charlotte Chaloner

Remarks on code availability

The code is available on a public Github repository, with a README and enough structure to follow what was done. The repo is well structured and simple enough to allow a user to review what precisely was done. No code or data for the neutralisation or the phylogenetic tree inference were provided, just the transmission model code and parameterisation.

We have updated the public GitHub repository to include analysis scripts for the neutralisation data and the BEAST input and output files used for phylogenetic tree inference. While the BEAST workflow itself is GUI-based and does not generate reproducible code, we have added a short tutorial in the repository README for constructing, running, and interpreting our sarbecovirus BEAST run.

Reviewer #2 (Remarks to the Author):

This manuscript presents a timely investigation into how post-pandemic population immunity—derived from both SARS-CoV-2 infection and vaccination—affects the emergence potential of zoonotic sarbecoviruses. A strength of the manuscript is how the authors combine *in vitro* neutralization assays (quantifying cross-immunity of SARS CoV-2 infection and vaccine immunity) with a thoughtfully constructed stochastic SEIRS model. With this they demonstrate that current levels of cross-immunity significantly reduce the likelihood of novel sarbecoviruses establishing sustained human transmission. Particularly commendable is the integration of empirical immunological data with dynamic modeling. This gives the study relevance to real-world pandemic preparedness. The exploration of both beneficial and potentially adverse effects of vaccination strategies adds nuance and depth to the findings. Overall, this study makes a valuable contribution to our understanding of SARS-CoV-X emergence in the post-COVID era and offers actionable insights for future vaccine development and surveillance strategies. I do have a few comments and concerns, as laid out below.

Comments

1. (Methods) Page 7 – It is unclear why the model incorporates age-specific vaccine effectiveness. While it is well-established that COVID-19 outcomes vary significantly by age, the evidence supporting meaningful variation in vaccine effectiveness across age groups is less familiar. Clarifying the rationale for this added complexity would help justify the model's structure.

We thank the reviewer for their comments, our approach to constructing our conceptual model of *Sarbecovirus* emergence was to include stratified effects – such as those describing age-specific vaccine effectiveness – where published data was available of adequate quality for parameterisation and the literature was in general consensus on the direction and magnitude of the effect. For example, we did not include age-stratified differences in incubation and recovery rates of SARS coronavirus infection as little consensus exists in the literature on host age differences in these infection

characteristics. In the case of age-specific vaccine effectiveness, multiple studies have quantified age-specific differences for COVID-19 vaccination, which we now include in the methods section describing age-stratified phenotypes. In these studies, vaccine effectiveness typically varied by ~8-10% from middle-aged individuals to elderly individuals, and a similar strength of effect can be seen in the publicly-available vaccine efficacy data we use to parameterise this age-stratified behaviour in our models (Supplementary Figure 11). As the reviewer suggests, it is likely that the inclusion of age-stratified vaccine effectiveness had only a minor influence on our model, and we do not expect it to have changed the overall direction of any effects we have reported. However, the age-stratified compartment structure was already present in our model to allow for other age-specific differences, and so the inclusion of age-specific differences in vaccine effectiveness required only a small amount of additional work to increase the realism of our model. To better highlight this, we have added the following to the methods section:

Materials & Methods (1.8.): “Evidence for age-stratified effects of vaccine-efficacy are also prevalent in the literature, although effect sizes are typically minor¹⁷⁻²².”

2. (Methods) Figure S2 – The reported one-dose COVID-19 vaccine coverage appears surprisingly low, never exceeding approximately 25%. This seems inconsistent with known vaccination rates in Scotland. Additionally, the model suggests that by 2022, nearly all individuals are categorized as "unvaccinated," which appears to result from a high rate of vaccine waning that outpaces new vaccinations. This assumption may substantially underestimate actual population immunity and warrants further explanation or validation.

Vaccination is represented in our model as a dynamic and multi-level process whereby individuals transition from an unvaccinated (U) phenotype to a single-dose protected (V_I), to a two-dose protected (V_{II}) phenotype, and regress to an unvaccinated phenotype at a daily probability calculated by applying an exponential decay model to antibody titre data [10.1172/JCI167955]. Figure S2 (now Figure S1) shows the percentage of the population expressing each vaccination phenotype over time, split by dose. The total percentage of the population protected by any dose (sum of V_s) can be more clearly seen in Figure S1B, and peaks at ~35%. The discrepancy between this value and reported vaccine coverage for Scotland is because the former is a transient phenotype that may be lost over time, while vaccine coverage is a history (i.e., has an individual ever been vaccinated – without reference to whether this occurred sufficiently long ago for this protection to have been lost). If we instead report vaccine coverage, our model closely resembles real-world values, with most age groups achieving coverage of > 90% by January 2022. We now include these data in Supplementary Figure 1C.

3. (Results) The results shown in Figure 4, where natural infection provides high cross-immunity but vaccines do not, appear more extreme and less plausible than those in Figure 3, where vaccine and infection-derived immunity are similar. It would be helpful to see intermediate scenarios—for example, where vaccine-derived immunity is half as effective as infection-derived immunity—to assess whether the conclusions hold under more moderate assumptions.

We agree with the reviewer that the scenario in Figure 4 (now Figure 5) – where vaccine derived immunity is considerably weaker than naturally derived immunity – is less plausible than other scenarios we have explored in this study. However, this could feasibly occur in scenarios where a highly strain-specific COVID-19 vaccine is used erroneously to resist the emergence of a novel *Sarbecovirus*. We include this result as it demonstrates that a detrimental effect of vaccination is theoretically possible when an antigenically overlapping virus is circulating in the human population. This effect disappears rapidly in less extreme scenarios, and it is essentially undetectable when vaccine-derived immunity is increased to be a third as effective as naturally derived immunity. We include this result as a new Supplementary Figure 7, with the following additions to the results:

Lines 240-245: “The detrimental effect of vaccination decreases rapidly as the levels of vaccine-cross protection increase; in scenarios where vaccine-cross protection was set to one-third the effectiveness of natural cross-protection, the effects of the vaccine campaign on natural and vaccine-derived population immunity become effectively balanced, producing little detectable positive or negative effect of a preventative vaccination campaign on the probability of SARS-CoV-X emergence (Supplementary Figure 7).”

4. (Results) While the authors explore a broad parameter space in their simulations, all scenarios focus on SARS-CoV-X emergence in 2024. Although the timing of vaccination relative to emergence is examined, it would strengthen the analysis to include simulations that assess risk over time. This is particularly relevant given the likely decline in population immunity as SARS-CoV-2 incidence remains low and vaccine uptake plateaus.

We agree with the reviewer that the emergence probability of SARS-CoV-X is likely to vary over time. Broadly, we expect that the emergence probability of a new *Sarbecovirus* will increase as population immunity decreases with time from the epidemic peaks of SARS-CoV-2 and the initial COVID-19 vaccine programs. Additionally, emergence probability is likely to have strong seasonal fluctuations from school holiday closures and environmental conditions that are known to be strong effectors of respiratory virus transmission. Our model has not been designed to explore these specific aspects of *Sarbecovirus* emergence – in essence it provides an estimate of the mean emergence probability across these effects – as it does not incorporate annual term-specific contact rates, does not include seasonal forcing of transmission rates, and, due to data constraints explained previously, simplifies post-Omicron SARS-CoV-2 prevalence to an equilibrium, which has the effect of also equalising population immunity over time. As such, we feel that an exploration of risk over time would be better explored with a model designed specifically to address this question. To highlight this as an important outstanding question we have added the following to the discussion:

Lines 332-335: “In addition, we do not explore here how temporal changes in population immunity, vaccine uptake, or seasonal variations in social contact behaviour or virus transmissibility may influence emergence risk over time, but these represent important extensions for future work.”

5. (Discussion) The authors acknowledge that their model assumes neutralizing antibody levels are a direct correlate of protection. However, this is a simplification, as immunity to SARS-CoV-2 involves multiple components, and protection against infection (which is most relevant for emergence risk) may differ from protection against disease. A more nuanced discussion of this limitation—and its implications for interpreting the model’s predictions—would be valuable.

We agree with the reviewer that this is a key assumption in our approach and have added the following to the discussion:

Lines 308-322: “Our study has various limitations. In the absence of more detailed information, we assumed that the *in vitro* activity of neutralizing antibodies directly correlates to *in vivo* protection from infection. Quantifying the exact shape of this relationship is challenging, requiring large longitudinal studies of animal or human infections (e.g. ⁴⁶⁻⁴⁸), and existing studies suggest this relationship varies across systems, populations, and study designs⁵. While neutralising antibodies represent a major component of the humoral response, protection *in vivo* also depends on non-neutralising antibody functions and cellular immunity, which may modify this relationship. In addition, although our data and model account for hybrid immunity, repeated exposures to antigenically distinct SARS-CoV-2 variants further modulate antibody breadth and potency.^{49,50} Because detailed infection histories were unavailable for our serum donors, the neutralisation values reported here likely reflect a mixture of singly and multiply exposed individuals and should therefore be viewed as population-weighted averages across varying exposure histories, rather than as discrete infection categories. To account for

these uncertainties, we also performed theoretical explorations in which cross-immunity and vaccine effectiveness were treated as independent of the neutralisation data. Overall, these simplifications reflect first-order approximations of complex immune processes that provide a framework for understanding emergence dynamics that may be refined in future work.”

Reviewer #3 (Remarks to the Author):

I enjoyed reading this manuscript, which tackles a genuinely difficult and policy-relevant question regarding how the accumulation of SARS-CoV-2 immunity might reshapes the landscape of future zoonotic sarbecovirus emergence. The study brings together neutralisation assays and an age-stratified SEIRS framework to estimate how existing cross-reactive immunity influences the probability that different coronaviruses might establish sustained human-to-human transmission. The question is timely, the experimental work is carefully executed, and the stochastic model (while necessarily stylised) offers useful qualitative insight into how vaccination strategies and cross-immunity interact to shape emergence risk. On the whole, this is a technically solid combination of laboratory measurements and stochastic epidemic modelling. With substantial revision it could, I believe, merit publication in Nature Communications. Below I lay out major points that require attention, followed by a set of more minor suggestions that are predominantly stylistic or more around clarifying points of uncertainty.

Major comments

1. Formal linkage between Fig. 1 and the model's parameterisation of immunological protection: The strength and novelty of the manuscript hinge on leveraging the neutralisation data to inform model parameters. At present, however, the manuscript does not explain in any transparent way how titres measured in Fig. 1 are converted into the cross-protection values that drive the results in Fig. 2D and Figs 3–4. Readers need to be able to trace that pathway explicitly. I therefore ask the authors to provide a clear mapping. My understanding (though it was hard to discern) is that the assumption was that it was a linear mapping – it's not clear to me how the absolute values used in this linear mapping were determined though. Moreover, previous work has shown a non-linear relationship, such as the framework used in Khoury et al. (Nat. Med. 2021) framework for scaling protection to a reference titre. This would perhaps be more appropriate to use, and either way, would benefit from a lot more description in the Methods and /or SI. A schematic or tabulated summary would be very helpful, and any arbitrary choices should be documented. More generally, I would urge the authors to anchor their neutralisation-derived efficacies to empirical estimates in the literature wherever feasible, especially given that much of these results hinge on estimates of the infection-blocking component of immunity, which was more modest than that for severe disease immunity.

We thank the reviewer for their comments and agree that it is important to make clear the assumptions linking the experimental neutralisation data in Figure 1 to the cross-protection values that influence our modelling results in Figure 2D and 3–4.

Both linear and non-linear relationships between *in vitro* neutralisation and *in vivo* protection from infection are frequently reported in the literature [recently reviewed in 10.1038/s41577-023-00858-w], and the shape of this relationship appears to be assay-, study-, and virus-dependent. The framework proposed in Khoury et al., (2021) uses normalisation to convalescent antibody titres to allow the authors to summarise across studies that have used different assays and approaches to measure neutralising antibody titre. The authors use the mean log-transformed fold-differences to convalescent titres of eight COVID-19 vaccines (analysed statistically as $n = 8$) to fit and report a non-linear relationship between these values and vaccine efficacy. We note, however, that the authors did not provide statistical justification for choosing a non-linear model to describe these data over a linear one.

We re-analysed their data (Figure 1A in Khoury et al., 2021) and found no statistical evidence to support a non-linear relationship over a linear relationship, as the addition of a non-linear term does not significantly increase the explanatory power of the model (see below). In our study, considerable individual and strain-level variation was observed in convalescent titres (Figure 1A), which may also suggest normalisation to convalescent titres is not an ideal approach to standardise across different neutralisation studies. For these reasons, we do not think that incorporating a framework such as the one proposed in Khoury et al., (2021) would be beneficial in this case.

Analysis of Variance Table

Model	Res.Df	RSS	DF	Sum of Sq.	F	P value
$y = \beta_0 + \beta_1x$	6	294.8				
$y = \beta_0 + \beta_1x + \beta_2x^2$	5	214.9	1	79.902	1.8591	0.2309

We agree with the reviewer’s important point that the mapping between *in vitro* neutralisation and *in vivo* protection is not likely to be 1:1 in reality, and this assumption may have shifted the exact values of emergence probability reported in our study (although it is unclear the direction in which they are likely to have been shifted, and we lack the additional evidence required to objectively move from this 1:1 assumption). Khoury et al., (2021), for example, estimate that an *in vivo* protection of 50% is achieved at titres ~20% of the mean convalescent titre, which may suggest that we are underestimating the cross-protection provided by our measured neutralisation titres. However, as explained above, directly linking the neutralisation values from Khoury et al., (2021) to our own study is not straightforward. These are key assumptions and discussion points for our model, and we make the following additions to the Methods, Results, and Discussion to clarify this to the reader:

Materials & Methods (1.7): “The proportion with which transmission rates are reduced in recovered and vaccinated individuals for SARS-CoV-X infections is assumed to correspond linearly (1-x) to the pseudotype neutralization data in Figure 1, and for SARS-CoV-2 re-infections from ³ (Supplementary Figure 9). The exact relationship between neutralization titres and protection from sarbecovirus infection is unknown, with both linear and non-linear relationships reported in the literature.¹⁵”

Lines 158-161: “To make the conversion from *in vitro* pseudotype neutralization to *in vivo* protection from infection, we have assumed that transmission probabilities scale by (1 - x) of cross-neutralisation, linearly reducing infection risk in individuals with higher immune cross-reactivity.”

Lines 308-322: “Our study has various limitations. In the absence of more detailed information, we assumed that the *in vitro* activity of neutralizing antibodies directly correlates to *in vivo* protection from infection. Quantifying the exact shape of this relationship is challenging, requiring large longitudinal studies of animal or human infections (e.g. ⁴⁶⁻⁴⁸), and existing studies suggest this relationship varies across systems, populations, and study designs⁵. While neutralising antibodies represent a major component of the humoral response, protection *in vivo* also depends on non-neutralising antibody functions and cellular immunity, which may modify this relationship. In addition, although our data and model account for hybrid immunity, repeated exposures to antigenically distinct SARS-CoV-2 variants further modulate antibody breadth and potency.^{49,50} Because detailed infection histories were unavailable for our serum donors, the neutralisation values reported here likely reflect a mixture of singly and multiply exposed individuals and should therefore be viewed as population-weighted averages across varying exposure histories, rather than as discrete infection categories. To account for these uncertainties, we also performed theoretical explorations in which cross-immunity and vaccine effectiveness were treated as independent of the neutralisation data. Overall, these simplifications reflect first-order approximations of complex immune processes that provide a framework for understanding emergence dynamics that may be refined in future work.”

2. Breadth of the neutralisation panel & sensitivity analysis: Only a small set of sarbecoviruses is assayed. This isn't necessarily an issue as the authors explore a range of assumptions around the degree of cross-protection in Figures 3 and 4, but I want to note that other laboratories have reported neutralisation against additional spike variants or coronaviruses (for example, Lei et al., Sci. Adv. 2023). It would greatly strengthen the analysis either to incorporate such published data (thereby expanding the antigenic landscape explored) or, if that is impractical, to compensate with a systematic sensitivity analysis that sweeps plausible ranges of cross-protection rather than fixing single values for each heat-map panel.

We agree with the reviewer that the inclusion of a more systematic analysis of cross-protection would increase the scope of our model estimates in Figure 2D (now Figure 3). Spike neutralisation assays are not standardised across the literature and normalising published results between studies is not straightforward (see previous comment). Additionally, a strength of our approach is the use of a biobank where neutralisation may be stratified by natural infection and vaccination history, but this information is also not consistently available across published datasets. In lieu of exploring additional real-world Sarbecoviruses, we have added two additional theoretical explorations of our immune parameters: an additional supplementary figure (Supplementary Figure 3) showing estimated probability of emergences over finer-scale values of natural-cross immunity and SARS-CoV-X R_0 , and a formal Sobol sensitivity analysis (Table 1) which includes variance component estimates of both natural and vaccine cross-immunity. These are highlighted with the following additions to the results:

Lines 215-218: “Emergence probability increased with R_0 and decreased with stronger cross-immunity (Figure 4B, Supplementary Figure 3), such that a hypothetical sarbecovirus with no immune cross reactivity and an R_0 of 6 had an estimated emergence probability of 61.08% (60.96-61.21%), decreasing to 17.99% (17.89-18.08%) as the level of cross reactivity approached 100%.”

Lines 189-195: “Sobol sensitivity analysis, performed on parameters for the R_0 of SARS-CoV-2 and SARS-CoV-X; the levels of natural immune protection; the waning rates of vaccine and natural immunity; and the hypothetical vaccination campaign, showed that most of the variation in the probability of SARS-CoV-X emergence was contributed to by the levels of natural cross-immunity and the R_0 of SARS-CoV-X (Table 1). Weaker effects on emergence probability were found for the R_0 of SARS-CoV-2, vaccine-related parameters, and the waning rate of SARS-CoV-X immunity, while no effect was detected for the waning rate of SARS-CoV-2 immunity across the parameter ranges tested.”

3. Characterising donor infection histories: Figure 1 is presented as a snapshot of neutralisation in individuals with “complex infection/vaccination histories”, yet the text does not state how those

histories were ascertained or whether sera derive from people with multiple SARS-CoV-2 exposures. Please clarify the inclusion criteria and laboratory confirmation of prior infection(s). More broadly, the Discussion should explicitly acknowledge that, as time passes, repeated and antigenically diverse SARS-CoV-2 exposures will further broaden population immunity. This potential broadening and strengthening of immunity following diverse repeated exposures isn't explicitly accounted for in the model currently if I have understood correctly (i.e. an individual's protection isn't explicitly a function of the entire pattern of their infection/vaccination history). It would be good to note this as a limitation, whilst also recognising that its inclusion would likely only have reinforced, rather than weakened, the manuscript's central conclusions around emergence odds.

To investigate the extent of pseudotype neutralisation in individuals with different histories of natural SARS-CoV-2 infection and vaccination, our study makes use of residual serum samples collected from primary and secondary care settings for the NHSGGC Biorepository from 31/03/2020 to 22/09/2021. Of the ~41,000 samples available from this repository, we characterise a representative subsample of 350 sera. This subsample was constructed through stratified random sampling to maintain the underlying demographic proportions of the full biobank, including age, sex, vaccination history and inferred infection history. Metadata accompanying these sera included the date of last PCR-confirmed positive infection with SARS-CoV-2, but not a complete history of infection. With this limited information, we have placed sera into categories of natural infection history based on the predominant SARS-CoV-2 variant at the time of their most recent PCR-positive test. This, by necessity, simplifies over the complexity of individuals with multiple variant exposures, which we agree has an established effect on the immunity expressed by individuals. Accounting for all possible patterns of multiple variant exposure explicitly in our model would add considerable complexity and also unique processes that we are unable to parameterize from the data and metadata available. As we apply no exclusion criteria for multiply exposed individuals, the levels of neutralisation we have measured should be interpreted as averages weighted by the proportion of singly and multiply exposed individuals in the Scottish population at the time of sample collection. To make this clear to the reader, we have added the following to the methods and discussion:

Materials & Methods (1.1.): "Of the approximately 41,000 samples collected, a representative subsample of 350 sera were selected that maintained the underlying demographic proportions of the full sample by stratified random sampling. Metadata on the complete infection history of each individual was unavailable, and so categories of natural infection history likely include both singly- and multiply-exposed individuals."

Lines 314-319: "In addition, although our data and model account for hybrid immunity, repeated exposures to antigenically distinct SARS-CoV-2 variants further modulate antibody breadth and potency.^{49,50} Because detailed infection histories were unavailable for our serum donors, the neutralisation values reported here likely reflect a mixture of singly and multiply exposed individuals and should therefore be viewed as population-weighted averages across varying exposure histories, rather than as discrete infection categories."

4. R_0 assumptions and calibration: The model assigns each coronavirus virus an R_0 taken from its nearest phylogenetic neighbour. This feels ad hoc, and the calibrated Wuhan- like value of 1.57 is lower than early-pandemic UK estimates (~3). I encourage the authors to (i) document the data, priors and goodness-of-fit for their calibration; and (ii) add a sensitivity analysis that spans a wider R_0 range for each coronavirus even if the central estimate remains phylogenetically informed.

We agree with the reviewer and have incorporated both of their suggestions into the manuscript. Our estimate of 1.57 for the R_0 of Wuhan has been returned twice: from our original grid-search process,

and now as the mean of the approximate posterior for this parameter from the more rigorous ABC calibration process. We now include a full summary of priors and approximate posteriors for this process in Supplementary Table 8. This R_0 value is on the lower end of what is reported in the literature for the Wuhan strain, however we note that estimates of R_0 vary considerably in the literature and are highly dataset, country, model, and context dependent. Indeed, these reasons are what prompted us to calibrate our model to the ONS infection survey dataset instead of relying on a fixed literature value for the R_0 of each SARS-CoV-2 strain. This calibration has produced a model with a stable Omicron equilibrium prevalence of ~2.2% after 2022, in line with current UK mean expectations over time, and it is this prevalence which most impacts the emergence probabilities of the novel Sarbecovirus introduced into the model in 2024. We now include a formal Sobol sensitivity analysis that quantifies the contribution of SARS-CoV-2 R_0 to variation in emergence probability. These results can be found in Table 1 and changes have been made to the results (see previous comment)

5. Early stochastic fade-out and model structure: Although the model is stochastic, it is spatially homogeneous and represents the population in aggregate. Real-world emergence is dominated by early, highly stochastic transmission chains in small clusters and many outbreaks will go extinct not because widespread transmission is not possible, but because of the inherent stochasticity of the early phases of emergence where the total number of infected individuals is small; a branching-process framework would likely represent that process more faithfully. Given the focus of the paper is on widespread emergence I do not think this would make a substantial difference and therefore don't consider a full re-implementation to be needed. However, the Discussion should acknowledge this limitation and note that the present framework probably overestimates of the absolute probability of emergence, whilst still serving an important role in identifying the relevant factors shaping emergence probability, and the relative probabilities across different scenarios.

We agree with the reviewer that this is an important simplification to acknowledge when modelling human populations using this approach. Our model exposes a single 30-year-old individual to SARS-CoV-X in each outbreak scenario, and this individual has a non-zero probability of contacting every other individual in the model each day they are infectious (as do all subsequent infectious individuals). These probabilities were calculated as mean age-stratified per-capita contact rates from the COMIX survey and follow-up study, which averages over individual-level differences and the branching processes underlying human social contact networks. In reality, social networks constrain contacts, making repeated contact with the same individuals highly likely (e.g., family members, work colleagues), and contact with socially distant individuals highly unlikely. The exposed individual's connectivity in the population will greatly impact the likelihood of novel virus emergence – for example we would expect an exposed 30-year-old teacher to have a much higher probability of producing an emergence event than non-teachers of the same age due to their close connection to school-age children, a high-contact rate group. Modelling approaches which incorporate social networks – such as individual-based models – are both computationally and developmentally expensive to implement, and we agree with the reviewer that a re-implementation is unnecessary to explore the questions of this study. To better acknowledge this limitation, we have added the following to the discussion:

Lines 327-332: “Our model also assumes a spatially homogeneous, fully mixed population and therefore does not fully capture the strong spatial stochastic effects that dominate during the earliest stages of emergence, when infections are few and often self-extinguishing. As a result, the present framework may modestly overestimate the absolute probability of emergence. Nonetheless, by focusing on the relative effects of key epidemiological and immunological factors, our analysis remains informative for understanding which parameters most strongly shape emergence risk.”

6. Vaccination-campaign scenarios: The main text outlines two “preventative” vaccination strategies. It was unclear to me the ways in which they differed – I didn’t understand them as described in the main text (nor the very limited additional description in the SI), nor whether either represents a campaign reactive to the detection of a novel coronavirus spill-over. I suggest:
 - overlaying illustrative incidence curves that depict the timing of doses under each scenario (perhaps as an inset to Fig. 2B–C);
 - clarifying in prose that one scenario reflects routine SARS-CoV-2 vaccination and the other a reactive roll-out in response to detection of SARS-X (or adding such a scenario if not already included); and
 - discussing how delays between first detection and mass vaccination might erode effectiveness.

My suggestions are therefore two fold: clarify extensively the vaccination scenarios, both graphically, and with prose. And then if the vaccination scenarios don’t include a reactive one currently, please add that in.

The preventative vaccination program in our model – which is in addition to routine SARS-CoV-2 vaccination through spring and winter boosters, which is present throughout all model runs in the manuscript – is controlled by three parameters: vaccine protection from SARS-CoV-X infection (0-1), vaccine program coverage of the human population (0-1) and the timing of the start of the vaccination program relative to the SARS-CoV-X exposure date (explored in the range -360 to +360 days). To clarify this graphically, we have added separate visualisations of the influence of each parameter on the model population in the new panels Figure 2D-F. Assuming SARS-CoV-X circulation is rapidly detected in the population, all vaccine programs with a positive value for timing can be considered reactive. Indeed, our model suggests that a rapid, high-coverage reactive response with a high cross-efficacy vaccine can reduce the emergence probability of a virus (Figure 4A), and that delays in detection and mass vaccination are likely to reduce the strength of this effect. To highlight this, the following has been added to the results and discussion:

Lines 181-186: “The model assessed the risk of a novel sarbecovirus emerging in humans under two scenarios: a) ongoing SARS-CoV-2 circulation with existing COVID-19 vaccination levels, and b) a hypothetical 2-month preventative vaccination campaign using current COVID-19 vaccines to resist SARS-CoV-X emergence. In the second scenario, the hypothetical vaccine campaign was allowed to vary in the timing of the start of the campaign relative to the date of SARS-CoV-X exposure (Figure 2D), the level of vaccine uptake (Figure 2E) and in the amount of cross-immune protection the vaccine provides from SARS-CoV-X infection (Figure 2F).”

Lines 347-345: “Moreover, our study suggests that the rapid and widespread deployment of readily available SARS-CoV-2 vaccines could be an effective strategy to limit the emergence of novel sarbecoviruses that aligns with the 100 Days Mission⁵⁹ to respond to future pandemic threats.”

Minor comments and stylistic suggestions

1. Figure 1: more detail in legend and methods required about how infection history was established and whether individuals with multiple variant exposures were included/excluded (and if so, how).
The legend for Figure 1 is already large due to the number of panels and information it provides, and so we have chosen to address this point in text through additions to the methods, results, and discussion (see previous comment).
2. Supplementary Information: expand to include full description vaccine-related parameters, in particular around how the results from Figure 1 were translated into estimates of immunity against infection. Also

clarify waning assumptions further and add sensitivity analyses exploring the impact of variation in waning on the results presented.

We have now included visualisations (Figure 2D-F) and full descriptions of our three vaccine-related parameters in the main and supplementary text. We have added an explanation to the results and expanded our explanation in the supplementary text describing the conversion from pseudotype neutralisation to protection from infection, highlighting the assumptions required for this translation. We have added to our description of the calculation of waning rates from literature data in the supplementary text and the assumptions used, and include our three waning rate parameters (SARS-CoV-2 waning rate, SARS-CoV-X waning rate, and vaccine waning rate) in our Sobol sensitivity analysis in Table 1. Additions to the text are as follows:

Lines 183-186: "...the vaccine campaign was allowed to vary in the timing of the start of the campaign relative to the date of SARS-CoV-X exposure (Figure 2D), the level of vaccine uptake (Figure 2E) and in the amount of cross-immune protection the vaccine provides from SARS-CoV-X infection (Figure 2F)."

Lines 192-195: "Weaker effects on emergence probability were found for the R_0 of SARS-CoV-2; the timing, coverage, and cross-protection of the preventative vaccine program; and the waning rate of SARS-CoV-X immunity, while no effect was detected for the waning rate of SARS-CoV-2 immunity across the parameter ranges tested."

Materials & Methods (1.7.): "The proportion with which transmission rates are reduced in recovered and vaccinated individuals for SARS-CoV-X infections is assumed to correspond linearly (1-x) to the pseudotype neutralization data in Figure 1, and for SARS-CoV-2 re-infections from 3 (Supplementary Figure 8). The exact relationship between neutralization titres and protection from sarbecovirus infection is unknown, with both linear and non-linear relationships reported in the literature."

3. Definition of emergence: I couldn't find a reference to this anywhere – how did the authors define emergence? Clarification of this is critically important.

We now also include this definition when we first mention emergence in both the results and discussion:

(Lines 152-153) "To assess the risk of zoonotic sarbecovirus emergence – defined here as a virus reaching endemicity after exposure to the human population – under varying levels of cross-protection..."

(Lines 258-261) "Here, we investigated the likelihood of a new zoonotic sarbecovirus emerging in the post-pandemic era – defined here as the virus reaching endemicity after exposure to the human population – given current levels of natural and vaccine-derived SARS-CoV-2 immunity."

4. Monte-Carlo noise in the heat-maps: In several panels adjacent tiles with nearly identical parameter values display markedly different colours, potentially indicative of stochastic variance in the results presented arising from not enough simulations having been run. If computationally feasible, please increase the number of stochastic realisations per parameter set.

We have attempted to minimise the stochastic noise that appears throughout the manuscript. Including the ABC calibration, Sobol sensitivity analysis, and all model figures, we have now run our model for ~248 CPU years, making use of the University of Glasgow MARS HPC cluster. In many cases, this represents a 25x increase in the number of iterations per parameter set over the original manuscript. Some stochastic noise remains but should no longer impact the clarity or interpretation of our model outputs.

I hope these comments are useful and congrats again on impressive and solid work. I look forward to seeing a revised version.

Remarks on code availability

I have not run the code but it's shared on Github which is best practice and gold standard for open-source code. I've reviewed it and looks good and applaud the authors for sharing transparently in this way.

Reviewer #4 (Remarks to the Author):

We hope this process has been helpful and encourage Nature Communications to continue their initiative.

Reviewer report for “Post-pandemic changes in population immunity have reduced the likelihood of emergence of zoonotic coronaviruses”

I enjoyed reading this manuscript, which tackles a genuinely difficult and policy-relevant question regarding how the accumulation of SARS-CoV-2 immunity might reshapes the landscape of future zoonotic sarbecovirus emergence. The study brings together neutralisation assays and an age-stratified SEIRS framework to estimate how existing cross-reactive immunity influences the probability that different coronaviruses might establish sustained human-to-human transmission. The question is timely, the experimental work is carefully executed, and the stochastic model (while necessarily stylised) offers useful qualitative insight into how vaccination strategies and cross-immunity interact to shape emergence risk. On the whole, this is a technically solid combination of laboratory measurements and stochastic epidemic modelling. With substantial revision it could, I believe, merit publication in *Nature Communications*. Below I lay out major points that require attention, followed by a set of more minor suggestions that are predominantly stylistic or more around clarifying points of uncertainty.

Major comments

2.1 Formal linkage between Fig. 1 and the model’s parameterisation of immunological protection: The strength and novelty of the manuscript hinge on leveraging the neutralisation data to inform model parameters. At present, however, the manuscript does not explain in any transparent way **how** titres measured in Fig. 1 are converted into the cross-protection values that drive the results in Fig. 2D and Figs 3–4. Readers need to be able to trace that pathway explicitly. I therefore ask the authors to provide a clear mapping. My understanding (though it was hard to discern) is that the assumption was that it was a linear mapping – it’s not clear to me how the absolute values used in this linear mapping were determined though. Moreover, previous work has shown a non-linear relationship, such as the framework used in Khoury *et al.* (Nat. Med. 2021) framework for scaling protection to a reference titre. This would perhaps be more appropriate to use, and either way, would benefit from a lot more description in the Methods and /or SI. A schematic or tabulated summary would be very helpful, and any arbitrary choices should be documented. More generally, I would urge the authors to anchor their neutralisation-derived efficacies to empirical estimates in the literature wherever feasible, especially given that much of these results hinge on estimates of the infection-blocking component of immunity, which was more modest than that for severe disease immunity.

2.2 Breadth of the neutralisation panel & sensitivity analysis: Only a small set of sarbecoviruses is assayed. This isn’t necessarily an issue as the authors explore a range of assumptions around the degree of cross-protection in Figures 3 and 4, but I want to note that other laboratories have reported neutralisation against additional spike variants or coronaviruses (for example, Lei *et al.*, *Sci. Adv.* 2023). It would greatly strengthen the analysis either to incorporate such published data (thereby expanding the antigenic landscape explored) or, if that is impractical, to compensate with a systematic sensitivity analysis that sweeps plausible ranges of cross-protection rather than fixing single values for each heat-map panel.

2.3 Characterising donor infection histories: Figure 1 is presented as a snapshot of neutralisation in individuals with “complex infection/vaccination histories”, yet the text does not state how those histories were ascertained or whether sera derive from people with multiple SARS-CoV-2 exposures. Please clarify the inclusion criteria and laboratory confirmation of prior infection(s). More broadly, the Discussion should explicitly acknowledge that, as time passes, repeated and antigenically diverse SARS-CoV-2 exposures will further broaden population immunity. This potential broadening and strengthening of immunity following

diverse repeated exposures isn't explicitly accounted for in the model currently if I have understood correctly (i.e. an individual's protection isn't explicitly a function of the entire pattern of their infection/vaccination history). It would be good to note this as a limitation, whilst also recognising that its inclusion would likely only have reinforced, rather than weakened, the manuscript's central conclusions around emergence odds.

2.4 R₀ assumptions and calibration: The model assigns each coronavirus virus an R₀ taken from its nearest phylogenetic neighbour. This feels ad hoc, and the calibrated Wuhan-like value of 1.57 is lower than early-pandemic UK estimates (~3). I encourage the authors to (i) document the data, priors and goodness-of-fit for their calibration; and (ii) add a sensitivity analysis that spans a wider R₀ range for each coronavirus even if the central estimate remains phylogenetically informed.

2.5 Early stochastic fade-out and model structure: Although the model is stochastic, it is spatially homogeneous and represents the population in aggregate. Real-world emergence is dominated by early, highly stochastic transmission chains in small clusters and many outbreaks will go extinct not because widespread transmission is not possible, but because of the inherent stochasticity of the early phases of emergence where the total number of infected individuals is small; a branching-process framework would likely represent that process more faithfully. Given the focus of the paper is on widespread emergence I do not think this would make a substantial difference and therefore don't consider a full re-implementation to be needed. However, the Discussion should acknowledge this limitation and note that the present framework probably overestimates of the absolute probability of emergence, whilst still serving an important role in identifying the relevant factors shaping emergence probability, and the relative probabilities across different scenarios.

2.8 Vaccination-campaign scenarios: The main text outlines two "preventative" vaccination strategies. It was unclear to me the ways in which they differed – I didn't understand them as described in the main text (nor the very limited additional description in the SI), nor whether either represents a campaign **reactive** to the detection of a novel coronavirus spill-over. I suggest:

- overlaying illustrative incidence curves that depict the timing of doses under each scenario (perhaps as an inset to Fig. 2B–C);
- clarifying in prose that one scenario reflects routine SARS-CoV-2 vaccination and the other a reactive roll-out in response to detection of SARS-X (or adding such a scenario if not already included); and
- discussing how delays between first detection and mass vaccination might erode effectiveness.

My suggestions are therefore two fold: clarify extensively the vaccination scenarios, both graphically, and with prose. And then if the vaccination scenarios don't include a reactive one currently, please add that in.

3 Minor comments and stylistic suggestions

- **Figure 1:** more detail in legend and methods required about how infection history was established and whether individuals with multiple variant exposures were included/excluded (and if so, how).
- **Supplementary Information:** expand to include full description vaccine-related parameters, in particular around how the results from Figure 1 were translated into estimates of immunity against infection. Also clarify waning assumptions further and

add sensitivity analyses exploring the impact of variation in waning on the results presented.

- **Definition of emergence:** I couldn't find a reference to this anywhere – how did the authors define emergence? Clarification of this is critically important.
- **Monte-Carlo noise in the heat-maps:** In several panels adjacent tiles with nearly identical parameter values display markedly different colours, potentially indicative of stochastic variance in the results presented arising from not enough simulations having been run. If computationally feasible, please increase the number of stochastic realisations per parameter set.

I hope these comments are useful and congrats again on impressive and solid work. I look forward to seeing a revised version.